

# Southern Ocean polynyas in CMIP6 models

Martin Mohrmann[1], Céline Heuzé[2], and Sebastiaan Swart[1, 3]

[1]Department of Marine Sciences, University of Gothenburg, Gothenburg, Sweden
[2]Department of Earth Sciences, University of Gothenburg, Gothenburg, Sweden
[3]Department of Oceanography, University of Cape Town, Rondebosch, South Africa

**Correspondence:** Martin Mohrmann (martin.mohrmann@gu.se)

**Abstract.** Polynyas facilitate air-sea fluxes, impacting climate-relevant properties such as sea ice formation and deep water production. Despite their importance, polynyas have been poorly represented in past generations of climate models. Here we present a method to track the presence, frequency and spatial distribution of polynyas in the Southern Ocean in 27 models participating in the Climate Model Intercomparison Project phase 6 (CMIP6) and two satellite based sea ice products. Only

half of the 27 models form open water polynyas (OWP), and most underestimate their area. As in satellite observations, three models show episodes of high OWP activity separated by decades of no OWPs, while other models unrealistically create OWPs nearly every year. The coastal polynya area in contrast is often overestimated, with the least accurate representations occurring in the models with the coarsest horizontal resolution. We show that the presence or absence of OWPs are linked to changes in the regional hydrography, specifically the linkages between polynya activity with deep water convection and/or the shoaling

of the upper water column thermocline. Models with an accurate Antarctic Circumpolar Current (ACC) transport and wind stress curl have too frequent OWPs. Biases in polynya representation continue to exist in climate models, which has an impact on the regional ocean circulation and ventilation that require to be addressed. However, emerging iceberg discharge schemes, vertical discretisation or overflow parameterisation, are anticipated to improve polynya representations and associated climate prediction in the future.

## 1 Introduction

Polynyas are areas of open water surrounded by sea ice. They are common features within the Southern Ocean winter sea ice, and are often classified into two different categories: coastal polynyas and open water polynyas (OWP). Coastal polynyas are usually latent heat polynyas that are kept open by winds that drive the sea ice away from the coastline or an obstacle, like an iceberg (Morales Maqueda et al., 2004). OWPs are kept open by thermodynamic processes, such as upwelling of sensible heat

or diffusive fluxes (Martinson and Iannuzzi, 1998) that transport warm water masses upwards, melting the sea ice and keeping the regional ocean ice-free. The heat source that keeps OWPs from freezing over is the presence of comparatively warm and salty Circumpolar Deep Water (CDW), which is usually located just below the base of the upper mixed layer (Santoso et al., 2006). These warmer waters transport heat from the depth to the surface by free or forced convection, caused by surface cooling / brine rejection, or wind and shear stresses respectively (Williams et al., 2007). While open-ocean deep convection is closely





linked to OWPs (Cheon and Gordon, 2019), deep convection is not a sufficient condition for OWPs to occur (Dufour et al., 2017).

Polynyas play a key role in sea ice and deep water formation. In winter, the strong temperature contrast between the open water in the polynyas and the cold Antarctic air causes significant heat loss of several hundred $W\ m^{-2}$ from the ocean to the atmosphere (Willmott et al., 2007). In fact, while the area of coastal polynyas is only about 1% of the sea ice area in the

30 Southern Ocean, 10% of Antarctic sea ice is produced there owing to this intense cooling. This rapid ice production in coastal polynyas is linked to intermediate water formation (Ohshima et al., 2016); it also produces a water mass of extremely high density, the Dense Shelf Water. This water is considered to be the main source of Antarctic Bottom Water, which forms the deepest layer of the global oceans (Orsi et al., 1999).

Some polynyas open at the same location every year (e.g. the Ross Sea Polynya), while others can be observed only once per

35 decade or less. The Weddell Sea Polynya has been the largest OWP to date and affected the water properties in the Weddell Sea for decades (Zanowski et al., 2015). It was first observed in the winters of 1974-1976 with an average area of up to $300\,000\ km^2$ (Carsey, 1980), produced about 5 Sv of Antarctic Bottom Water (Wang et al, 2017), and led to a significant cooling at mid depth (Cheon and Gordon, 2019). More than forty years later, in 2016 and 2017, the Maud Rise polynya re-opened (Swart et al., 2018; Heuzé et al., 2019; Francis et al., 2019) and reached an area larger than $50\,000\ km^2$, similar to the polynya of

40 October-November 1973 (Cheon and Gordon, 2019). The presence of OWPs is correlated to a positive Southern Annular Mode (SAM) and strong Southern Hemisphere westerlies (Campbell et al., 2019; Cheon et al., 2014). Behrens et al. (2016) also found a positive correlation of open ocean convection, as observed in OWPs, with the Antarctic Circumpolar Current (ACC) transport on multidecadal time scales.

Polynya detection in observational data (e.g. Markus and Burns, 1995; Kern et al., 2007; Ohshima et al., 2016) and ocean re-

45 analysis products (Aguiar et al., 2017) and the role of open water polynyas in spurious deep water formation in CMIP5 (Heuzé, 2015) have been discussed before, but despite the polynyas' crucial role for sea ice production and deep water properties, an evaluation and comparison of their representation across current climate models is missing.

Here we determine the characteristics, causes and impacts of Southern Ocean polynyas in 27 models that participated in the Climate Model Intercomparison Project phase 6 (CMIP6, Eyring et al., 2016). We present a new method to detect polynyas

from daily and monthly sea ice concentration or thickness in CMIP6 models and observational data sets in Section 3. We use this approach to assess Southern Ocean polynya characteristics (spatial distribution, frequency and seasonality) for the entire CMIP6 historical run and compare it with observational satellite data when available in Section 4. As observational data from within open water polynyas are very sparse, we continue with a comparison of the models' hydrography in sea ice covered conditions to that occurring during OWP events in order to assess OWPs' impact on the entire water column. We also analyse

indicators of deep convection and upwelling as possible causes for the polynya formation and maintenance in CMIP6 models. Finally, with the obtained polynya statistics from the CMIP6 models, we determine whether the observed correlations between the SAM, Southern Hemisphere westerlies, ACC transport and the presence or magnitude of OWPs hold true across CMIP6 models (Section 5). We first consider the entire Southern Ocean south of 55°S, and then focus on the Weddell Sea region, between 65°W and 30°E.



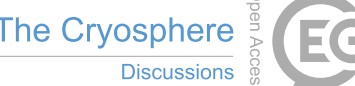

**Table 1.** The 27 CMIP6 models used in this study. Columns show the model names, ocean component, nominal horizontal resolution of the ocean ($R_o$), atmosphere component, horizontal resolution of the atmosphere ($R_a$), sea ice component, vertical discretisation scheme including number of vertical levels (z: depth-level, $\rho$: isopycnal, $\sigma$: terrain-following, several symbols: hybrid grid), data availability of monthly sea ice concentration ($C_m$), daily sea ice concentration ($C_d$), monthly sea ice thickness ($T_m$) and daily sea ice floe thickness ($FT_d$).

| Model | Ocean comp. | $R_o$ | Atmos comp. | $R_a$ | Sea Ice comp. | Vertical | $C_m$ | $C_d$ | $T_m$ | $TF_d$ |
|---|---|---|---|---|---|---|---|---|---|---|
| ACCESS-CM2 | ACCESS-OM2 | 100 | MetUM-HadG. | 250 | CICE5.1.2 | z* 50 | ✓ | ✓ | ✓ | ✓ |
| ACCESS-ESM1-5 | ACCESS-OM2 | 100 | HadGAM2 | 250 | CICE4.1 | z* 50 | ✓ | | ✓ | |
| BCC-CSM2-MR | MOM4 | 50 | BCC AGCM3 | 100 | SIS2 | z 40 | ✓ | ✓ | ✓ | ✓ |
| BCC-ESM1 | MOM4 | 50 | BCC AGCM3 | 250 | SIS2 | z 40 | ✓ | ✓ | ✓ | ✓ |
| CAMS-CSM1-0 | MOM4 | 100 | ECHAM5 | 100 | SIS 1.0 | z 50 | ✓ | | | |
| CanESM5 | NEMO3.4.1 | 100 | CanAM5 | 500 | LIM2 | z 45 | ✓ | ✓ | ✓ | ✓ |
| CESM2 | POP2 | 100 | CAM6 | 100 | CICE5.1 | z 60 | ✓ | ✓ | ✓ | ✓ |
| CESM2-FV2 | POP2 | 100 | CAM6 | 250 | CICE5.1 | z 60 | ✓ | ✓ | ✓ | ✓ |
| CESM2-WACCM | POP2 | 100 | WACCM6 | 100 | CICE5.1 | z 60 | ✓ | ✓ | ✓ | ✓ |
| CESM2-WACCM-FV2 | POP2 | 100 | WACCM6 | 250 | CICE5.1 | z 60 | ✓ | ✓ | ✓ | ✓ |
| CNRM-CM6-1 | Nemo | 100 | Arpege 6.3 | 250 | Gelato 6.1 | z* 75 | ✓ | ✓ | ✓ | ✓ |
| CNRM-ESM2-1 | Nemo | 100 | Arpege 6.3 | 250 | Gelato 6.1 | z* 75 | ✓ | ✓ | ✓ | ✓ |
| EC-Earth3 | NEMO3.6 | 100 | IFS cy36r4 | 100 | LIM3 | z*75 | ✓ | ✓ | ✓ | ✓ |
| EC-Earth3-Veg | NEMO3.6 | 100 | IFS cy36r4 | 100 | LIM3 | z* 75 | ✓ | ✓ | ✓ | ✓ |
| GFDL-CM4 | MOM6 | 25 | GFDL-AM4.0 | 100 | GFDL-SIM4p | $\rho$ - z* 75 | ✓ | | ✓ | ✓ |
| GFDL-ESM4 | MOM6 | 50 | GFDL-AM4.1 | 100 | GFDL-SIM4p | $\rho$ - z* 75 | ✓ | | ✓ | |
| HadGEM3-GC31-LL | NEMO | 100 | MetUM-HadG. | 250 | CICE-HadGEM | z* 75 | ✓ | | ✓ | |
| IPSL-CM6A-LR | NEMO-OPA | 100 | LMDZ (NPv6) | 250 | NEMO-LIM3 | z* 75 | ✓ | ✓ | ✓ | ✓ |
| MIROC6 | COCO4.9 | 100 | CCSR AGCM | 250 | COCO4.9 | z - $\sigma$ 62 | ✓ | | | ✓ |
| MIROC-ES2L | COCO4.9 | 100 | CCSR AGCM | 500 | COCO4.9 | z - $\sigma$ 62 | ✓ | | | ✓ |
| MPI-ESM-1-2-HAM | MPIOM1.63 | 250 | ECHAM6.3 | 250 | unnamed | z 40 | ✓ | ✓ | ✓ | ✓ |
| MPI-ESM1-2-HR | MPIOM1.63 | 50 | ECHAM6.3 | 100 | unnamed | z 40 | ✓ | ✓ | ✓ | ✓ |
| MPI-ESM1-2-LR | MPIOM1.63 | 250 | ECHAM6.3 | 250 | unnamed | z 40 | ✓ | ✓ | ✓ | ✓ |
| MRI-ESM2-0 | MRI.COM4.4 | 100 | MRI-AGCM3.5 | 100 | MRI.COM4.4 | z* 60 | ✓ | ✓ | ✓ | ✓ |
| NorCPM1 | MICOM1.1 | 100 | CAM-OSLO4.1 | 250 | CICE4 | z - $\rho$ 53 | ✓ | | | |
| SAM0-UNICON | POP2 | 100 | CAM5.3 | 100 | CICE4.0 | z 60 | ✓ | ✓ | ✓ | ✓ |
| UKESM1-0-LL | NEMO-HadG. | 100 | MetUM-HadG. | 250 | CICE-HadGEM | z* 75 | ✓ | | ✓ | |



## 2  CMIP6 output fields and observational data

We start with a description of the analysed data and then present the algorithm that we use in Section 3 to find polynyas. We used for this study the daily sea ice concentration from OSI-450 (Lavergne et al., 2019), the daily sea ice SMOS thin ice thickness (Huntemann et al., 2014) and the sea ice output from the historical run of 27 CMIP6 models (Eyring et al., 2016, listed in Table 1). We created this subset of all available CMIP6 models by filtering for models that participate in the 'historical'
CMIP6 scenario, have at least one sea ice variable downloadable in a monthly or daily format and provide adequate projection and cell area information. The historical CMIP6 run covers the years from 1st January 1850 to 31st December 2014 and was forced with observed historical greenhouse gas concentrations. As of the latest date of download (October 2020), only one ensemble member was available for the majority of models. Consequently, we chose one representative ensemble member (r1i1p1f2 for CNRM models, r1i1p1f3 for HadGEM, r1i1p1f1 otherwise) for each model. This way, we ensure consistency
with other CMIP sea ice area evaluations (Roach et al., 2020; Turner et al., 2013). For the assessment of polynya activity in the Southern Ocean, we use sea ice concentration and sea ice thickness data. In the 27 analysed models, the nominal horizontal resolution varies between 0.25° and 1.5° for the ocean (including the sea ice) and between 1° and 2.5° for the atmosphere (Table 1). All computations were performed on the models' native grid unless specified otherwise. The grid cell area ('areacello') was used to compute the surface area of sea ice and polynyas. The models were purposely not detrended, as we want to determine
the accuracy of their historical run with ongoing climate change incorporated into the analysis. For the assessment of polynya activity in the Southern Ocean, we use the sea ice concentration and sea ice thickness as discussed below.

### 2.1  Sea Ice Concentration

Sea ice concentration (CMIP6 parameter: 'siconc') is available at a daily resolution for only 18 of the 27 models, whereas it is available at a monthly resolution for all 27 models (Table 1). Furthermore, the daily sea ice concentration is not available
for the full historical period for the models MRI-ESM2-0 (1st January 1920 - 31st December 2015) and SAM0-UNICON (1st January 1950 - 31st December 2015). Routine satellite based sea ice concentration observations are available from January 1979 onwards, so there are 35 years of overlap with the historical CMIP6 model run on which we can perform our comparisons.

   We use the sea ice concentration product OSI-450 (Lavergne et al., 2019) for the time period 1979-2015. For the time after 2015, an extension with the name OSI-430-b is available that is processed with the same algorithms (Lavergne et al., 2019).
OSI-450/430b are sea ice concentration products, computed from SMMR (1979-1987), SSM/I (1987-2008), and SSMIS (2006-2020) microwave radiometers and ECMWF-ERA-Interim reanalysis data (Dee et al., 2011). It is provided at a 25 km x 25 km horizontal resolution and daily temporal resolution from 1st January 1979 to 2020 onwards (every other day until 1985).

   Using sea ice concentration to detect polynya activity has the advantage that it is available for the largest number of CMIP6 models (Table 1) and has been observed by satellite for more than 40 years. The disadvantage is that sea ice concentration is
a poor choice to detect coastal polynyas that are often covered by newly forming thin sea ice in winter and their full extent is better characterised by a low sea ice thickness (Ohshima et al., 2016). Polynya detection by sea ice thickness thresholds is well





established (e.g. Ohshima et al., 2016; Nakata et al., 2015; Kern et al., 2007) and detects even polynyas that are covered by a thin layer of newly formed sea ice.

## 2.2 Sea Ice Thickness

For the CMIP models, two output variables related to the sea ice thickness are available. The sea ice volume per grid cell area ('sivol') has been available in earlier CMIP versions, and is commonly and hereafter referred to as equivalent sea ice thickness or just thickness. With CMIP6, a new sea ice variable called ice floe thickness ('sithick') is introduced and available for the majority of models (Table 1). The equivalent sea ice thickness sivol is available for 23 of the models, but only at a monthly averaged resolution. Polynyas are dynamic processes, and can change their area and position drastically within few

100 days (Nakata et al., 2015; Kern et al., 2007); thus it would be optimal to analyse daily data from all data sources. The ice floe thickness sithick is available on a daily resolution for 21 of our models, but we found it not good for our polynya detection, as we show in Section 3.3.

For observational comparison of the sea ice thickness, we use the satellite-based SMOS thin sea ice thickness product (Huntemann et al., 2014) at a daily resolution. The product consists of maps of sea ice thickness up to 0.5 m, derived from the

105 satellite-borne L-band radiometer SMOS. The data are distributed at a horizontal resolution of 12.5 km x 12.5 km and available from 1st June 2010 onwards. The sea ice thickness calculation from satellite data is a relatively new method that works best for ice thickness values of up to 50 cm, with an average retrieval error of about 30% of the retrieved value (Huntemann et al., 2014). It does not take differences in sea ice concentration into account and might thus be biased in regions with low sea ice concentration. The accuracy of the method is negatively affected by melt ponds (ponds that the form on the sea ice in the

110 melting season), but low air humidity causes melt ponds to occur much less frequently on the Antarctic sea ice compared to the Arctic sea ice (Andreas and Ackley, 1982). The thin sea ice retrieval is a relatively new method that is more established for Arctic regions (e.g. Tietsche et al., 2018), but papers about its quality and applicability for the Southern Ocean are in preparation (Mchedlishvili et al., 2021; Tian-Kunze et al., 2021). Even though the 10 years time period is too short for deriving climatological mean values, we decided to include it as an observational baseline, where relevant, to compare the CMIP6 data

to.

Due to the described advantages and disadvantages of using either sea ice concentration (siconc) or thickness (sivol), we have decided, where practical, to analyse both and present the resulting data side by side in sections 3 and 4.

## 2.3 Vertical Ocean Profiles

We do not limit our analysis to the surface of the ocean, but also analyse vertical ocean profiles within CMIP6 polynyas. The

120 majority of models use a z-level vertical grid in the ocean, with the exceptions of GFDL-CM4/ESM4, MIROC6/ES2L and NorCPM1. The GFDL models use isopycnal coordinates in the interior ocean and rescaled geopotential vertical coordinates in the mixed layer. The MIROC6 model uses hybrid z - sigma coordinates between the sea surface and a fixed geopotential depth and a z-level grid below. NorCPM1 uses isopycnal coordinates in the interior ocean and z coordinates in the mixed layer; it is also the only model using data assimilation (Counillon et al., 2016). To investigate the hydrography we extracted vertical



profiles from the monthly ocean salinity ('so') and potential temperature ('thetao'). For observational comparison of the water properties, data from a SOCCOM vertical profiling float (Johnson et al., 2018) is used. We use temperature and salinity profiles measured by the SOCCOM float 5904471, because it provides a long record of profiles including rare vertical profiles from within the 2017 Maud Rise OWP (Campbell et al., 2019).

## 3 Polynya detection

In this work, we present a new method to detect and distinguish coastal and open water polynyas using either sea ice concentration or equivalent sea ice thickness (Fig. 1). We apply this method to the observational, satellite-based products and the CMIP6 models listed in Table 1.

### 3.1 Our algorithm to detect polynyas

With the aim of detecting polynyas, we start with the sea ice concentration or equivalent thickness (Fig. 1a). A "flood-fill"
algorithm masks out the open ocean beyond the northern sea ice extent (Fig. 1b), after which a threshold filter returns all grid cells that are classified as "polynyas" (Fig. 1c). To differentiate coastal polynyas from OWPs, in a third step the flood fill algorithm is applied to the Antarctic continent (Fig. 1d). Coastal polynyas, adjacent to the land, are hence also covered by the flood fill algorithm. The grid cells below the threshold that remain are classified as "OWP". The polynya area is the sum of the grid cell areas. The obtained OWP areas can be subtracted from the total polynya area to obtain the coastal polynya areas.

Depending on the chosen threshold and sea ice variable, the resulting polynya and sea ice areas differ. For example a closed layer of thin sea ice will be detected as a polynya if using the threshold thickness criteria but not by its sea ice concentration; an area with relatively sparse but thick sea ice will be classified as a polynya by the threshold concentration (siconc) criteria but not by its sea ice thickness (sivol). In the literature, a wide range of values for the concentration thresholds are in use. Previous studies on polynyas tend to choose a high sea ice concentration threshold to maximise the number of polynyas detected (e.g.,
Arbetter et al., 2004; Gordon et al., 2007), while sea ice-specific studies choose a low threshold to capture all areas with sea ice. For example, Shu et al. (2015); Beadling et al. (2020) use a 15% threshold to estimate the sea ice extent in CMIP5 and CMIP6 models, Kern et al. (2007) classify areas with up to 45% sea ice coverage as open water areas, while Smedsrud (2005) points out that sea ice coverages of 80-90% already allow for significant heat exchange and can therefore be classified as "polynyas". The influence of different sea ice thresholds on our four step algorithm is visualised in Fig. 2: the areas that are classified as
polynyas increase with higher sea ice concentration thresholds, until the first polynya areas merge with the open ocean (Fig. 2b,e). High sea ice concentration thresholds maximize the number of detectable coastal polynyas, but lead to poor recognition of OWPs (Fig. 2e). The higher the chosen sea ice threshold, the lower the area that is classified as sea ice. We found that almost no coastal polynyas are detected at a 15% threshold in our observational product (Fig. 2b,c) and most CMIP6 models (not shown). Therefore, we chose a 30% sea ice concentration threshold as a good compromise: higher thresholds lead to a
strong negative bias in sea ice area compared to other papers and opens polynyas to the open ocean; lower values leave too many polynyas undetected. For the sea ice thickness threshold, we chose a value of 12 cm. This is consistent with Nakata et al.





**Figure 1.** A four step algorithm classifies surface type areas in the marginal ice zone. **(a)** August 2002 monthly mean sea ice concentration from ACCESS-ESM1.5. **(b)** Sea ice map after covering the open ocean with the flood fill algorithm. **(c)** Application of a threshold filter, highlighting the total polynya areas. **(d)** Flood fill algorithm is applied to Antarctic fast land, the remaining open water or low sea ice areas are classified OWPs.

(2015), who give the total polynya extent as the areas occupied by open water and thin ice (up to 12 cm). For a discussion of the effect of different sea ice thickness thresholds, see Nakata et al. (2015) and Kern et al. (2007).

Some CMIP6 models provide only monthly averaged data, others include the sea ice concentration at daily resolution (Table 1). It is obvious that monthly data is unsuitable to detect a phenomenon that may last but a few days, and that consequently



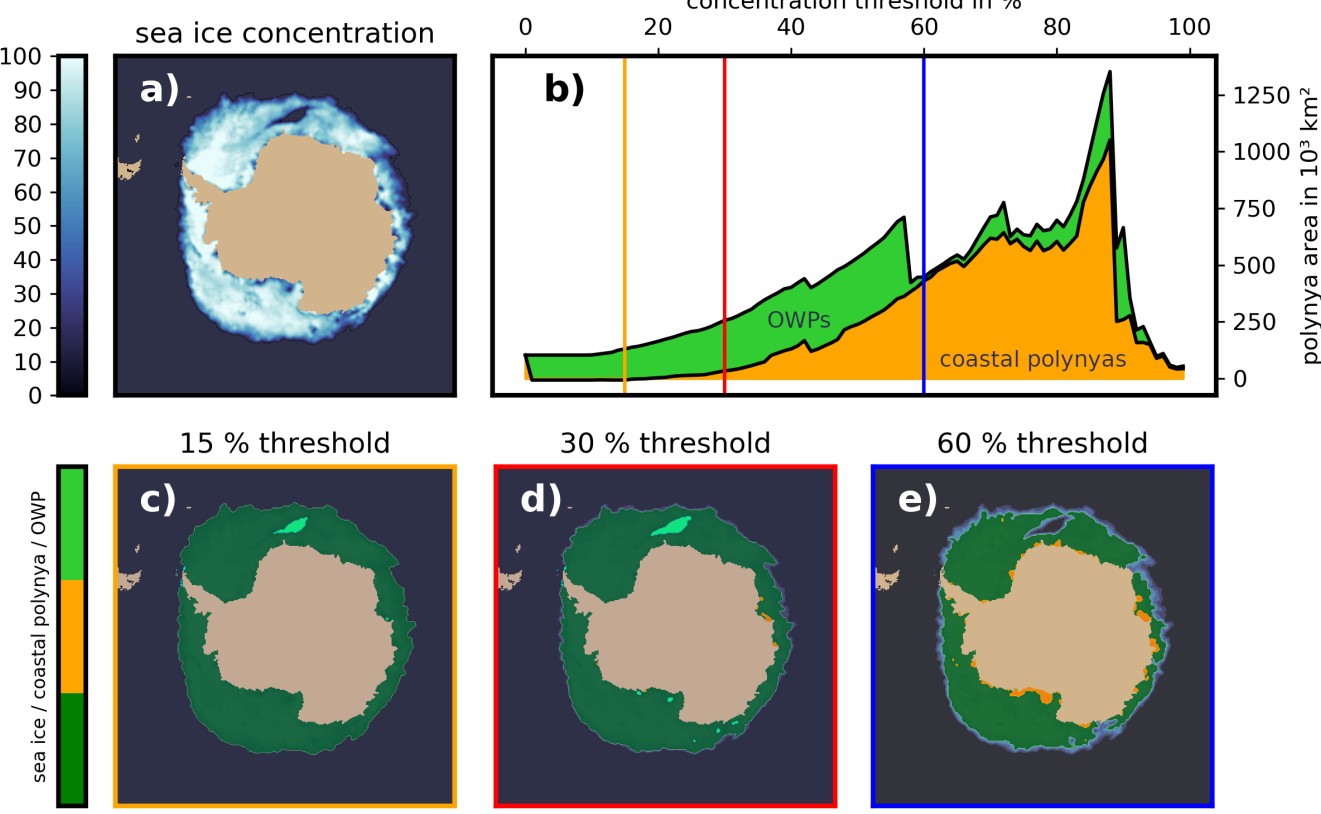

**Figure 2.** Illustration of the effect of different sea ice concentration thresholds on the detection of polynyas. **(a)** Observational sea ice concentration (OSI450) from 25th November 2017. **(b)** Combined area of coastal polynyas (orange) and OWPs (green) for sea ice concentration thresholds from 0 to 100%. The vertical lines correspond to the results of our algorithm for **(c)** 15%, **(d)** 30%, and **(e)** 60% sea ice threshold.

the usage of monthly averaged sea ice output may affect the accuracy of the mean polynya areas. We transformed the data to a common time resolution to allow for a comparison. Here we used yearly polynya area averages:

$$\bar{A}_y(\text{daily data}) = \frac{\sum\limits_{1.May}^{15.Nov} A_{\mathrm{p}}(t)}{N_{\text{days}}}, \quad \bar{A}_y(\text{monthly data}) = \frac{\sum\limits_{May}^{Nov} A_{\mathrm{m}}(t)}{N_{\text{months}}}. \tag{1}$$

As we show in Section 4.2, polynyas grow very large in summer season, but we focus on winter polynyas here due to their

large importance in ocean-atmosphere heat exchange and sea ice formation. We used the results from Equation (1) to create scatter plots with least squares linear regressions (Fig. A1, A2). For the observational sea ice concentration, the slope of the regression fit is 0.73, indicating that total polynya areas are underestimated by 27% if computed from monthly averages and should be derived from daily data wherever possible. The regression is worst with OWP, probably because of the small number of OWPs in the observational data (Fig. A1). Seven of 18 CMIP6 models have a regression slope higher or equal to the

observational product (ACCESS-CM2, BCC-CSM2-MR, CNRM-CM6-1, CNRM-ESM2-1, EC-Earth3(-Veg), MPI-ESM1-2-





HR), i.e. a better agreement between daily and monthly polynya areas, while the rest have a lower slope. We find a higher correlation for models including larger polynyas (total polynya area $> 100\text{x}10^3\text{km}^3$) compared to models that have only minor polynyas. We suspect that smaller polynyas are more prone to relative changes in area due to ice drift.

The correlation of yearly polynya areas from equivalent sea ice thickness (sivol) with sea ice concentration (siconc) shows a slope of 0.31 for the observational data sets SMOS thin ice and OSI-450/430 (Figure A2). Here we could only use the common time period from 2010 to 2020. The low slope indicates that our algorithm classifies 69% less area as polynya when using sea ice concentration. This is not necessarily an erratic underestimation in this case, but rather originates in different definitions of a polynya either using a concentration or thickness threshold. However, for 12 of the 23 models this slope is higher, indicating a closer relationship between low sea ice thickness and concentration. We found low slopes in models with more coastal polynyas

(e.g. MRI-ESM2.0: 0.00, IPSL-CM6A: 0.05, CESM2: 0.11), but also in the BCC models that form wide areas of thin sea ice (<10 cm) followed by a thicker ring along the outer sea ice edge (not shown).

### 3.2 Averaging methods

For readability, we condensed the data further in several ways depending on the objective of the analysis. To analyse the spatial distribution of polynyas, we present sea ice maps for September with polynya occurrences highlighted (Fig. 3, A3, A4). We

chose the month of September as it typically features the austral sea ice maximum and is therefore discussed in related CMIP6 sea ice and Southern Ocean evaluations (e.g., Beadling et al., 2020; Roach et al., 2020). However, we also give the different seasonality of the modelled coastal and open water polynyas in section 4.2. In Figures 6, 7, A1, A2 and for the polynya areas in Table 2, we used equation 1 first and took the mean over all model years then. For the sea ice extent in Table 2 and the maximum polynya extents in Fig. 6, we used

$$\bar{A} = \frac{\sum\limits_{\text{years}} \max{(A_{\text{p}}(t))}\Big|_{1.\text{May}}^{15.\text{November}}}{N_{\text{years}}}, \quad \bar{A} = \frac{\sum\limits_{\text{years}} \max{(A_{\text{p}}(t))}\Big|_{\text{May}}^{\text{November}}}{N_{\text{years}}} \tag{2}$$

for daily and monthly variables respectively. For the figures within this paper we present the result of the analysis of either the sea ice concentration or thickness in either daily or monthly resolution to keep the data comparable. Not all plots include all 27 analysed models, since some models do not provide all output variables (Table 1). To avoid unnecessary repetitions in the paper, we do not always present the results from all sea ice variables used, but keep some results exclusive to the supplementary

figures.

In Section 4, we limit the analysis to the period May to November in order to filter out summer polynyas. Throughout the paper, we use the mean yearly polynya area for the winter season if not specified otherwise. In sea ice melting summer conditions, polynyas become often very large, but we want to focus on winter polynyas due to their importance in heat exchange and ice and deep water formation. Moreover, it is not recommended to use the SMOS thin ice product during melting season

(Huntemann et al., 2014). When comparing model output data to the observational sea ice concentrations, we further limit the yearly mean values to a common time period from 1979-2015 to ensure consistency.



### 3.3 Sea ice floe thickness

The sea ice floe thickness (sithick) is defined as the actual thickness of the sea ice in the CMIP6 models, averaged over the ice-covered part of the grid cell (Huntemann et al., 2014). While the detected polynya locations and areas in our analysis of

205 the sea ice concentration and thickness (Fig. 3, A3, A4) agree reasonably well with observations for most models and will be discussed further, we could not achieve satisfactory results if using only the sea ice floe thickness as input data. The majority of the models (e.g. ACCESS, BCCs, CESM2s, MPIs, IPSL-CM6A-LR, MRI-ESM2-0, SAM0-UNICON) included increasing sea ice floe thickness beyond the outer sea ice extent of the concentration maps. These models also have a considerable mean floe thickness inside areas that would be classified as polynyas by low sea ice concentration and equivalent thickness (Fig. 4).

A combined approach in which either low sea ice concentration or thickness or the product of both classifies an area as polynya would be possible, but out of the 21 models that include daily sea ice thickness, three were lacking daily sea ice concentration. Where daily data is provided, it is sometimes only available for a limited period ( MRI-ESM2-0, SAM0-UNICON). Moreover, we found that the MPI models show a strange behaviour, where the floe thickness is either >0.5m or zero, but never takes values <0.5m. We assume that the differences in implementation of the floe thickness output variable are due to the

novelty of this variable with CMIP6, so that no conventions over the exact interpretation and implementation of the variable have formed yet. We will focus instead on the equivalent sea ice thickness (sivol) and concentration (siconc) variables for the rest of the paper.

### 3.4 Caveat

To asses and compare polynya activity in a large number of models, we analyzed different output variables for equivalent sea

ice thickness and concentration. The equivalent sea ice thickness 'sivol' was only available at a monthly resolution. We hope to see more daily sea ice data for all models published in the next CMIP iteration.

The current algorithm differentiates coastal and open ocean polynyas by their direct connection to the continent. In general, we find a good agreement between the algorithm's classification and our visual validation. However, occasionally the algorithm classifies a polynya that is (at least partially) located on the coastal shelf as an open water polynya; or a large open water

polynya neighbouring land on one grid cell, as a coastal polynya. While this classification holds true for a strict definition of the polynyas, a polynya on the coastal shelf is unlikely to be a sensible heat polynya driven by deep water convection. We have seen this for the model IPSL-CM6A-LR that was reported to undergo deep convection (Heuzé, 2021), but shows little OWPs in our analyses. Looking at the sea ice distribution, the model has large polynyas that are adjacent to the coastline but span hundreds of kilometers into the ocean. Even though those are strictly speaking coastal polynyas and classified as such by our

algorithm, the underlying physical process is likely deep water convection. An improvement of our method could be achieved by taking the water depth at each grid cell into account in the classification algorithm. Thus polynyas located on the shelf could be classified as coastal polynyas, while coastal polynyas extending further into the open ocean could be (partially) classified as open water polynyas.





The resulting polynya areas derived from observational and model sea ice thickness are not directly comparable for two
reasons: The historical CMIP6 run (1st January 1850 - 31st December 2014) and the SMOS thin ice thickness data (1st June
2010 onwards) have less than five years of overlap. The second reason is their different definitions. The CMIP6 equivalent
thickness variable describes the sea ice volume, or more exactly the sea ice mass per grid cell area (Notz et al., 2016), while
the SMOS thin ice thickness describes the physical sea ice thickness (Huntemann et al., 2014).

In several models (GFDL-, ACCESS-, UKESM1-0-LL) multi-centennial variability causes periods with high polynya ac-
tivity followed by times with few polynyas (Sellar et al., 2019). The ACCESS-ESM1-5 shows a long term warming trend of
more than 1°C over the 165 model years (not shown). The BCC models have a strong multidecadal variance in sea ice extent
(Beadling et al., 2020) which also affects the polynya areas (Section 4.2). In these models, the results for the polynya activity
may vary depending on the chosen ensemble member and time period.

## 4   Polynya statistics in CMIP6

The aim of our study is to determine whether the representation of polynyas in CMIP6 models is accurate in terms of location
(section 4.1) and frequency (section 4.2). To do so, we evaluate the observational products and the model output with the same
methods introduced in Section 3. We finally investigate the effect of the modelled polynyas on modelled stratification in section
4.3.

### 4.1   Spatial distribution of Southern Ocean polynyas

We here evaluate the modelled location and spatial extent of coastal and open water polynyas from the sea ice concentration
and thickness variables. All 27 CMIP6 models exhibit coastal polynyas. The largest coastal polynya, the Ross Sea polynya, is
represented in all models (Fig. 3, A3, A4), although some models open the Ross Sea polynya later than September. Smaller
polynyas, usually located by bays, headlands or islands along the continent are best reproduced by models with a high horizon-
tal resolution (Fig. 3, A3, A4). The GFDL-CM4 model with the highest horizontal resolution (ca. 0.25°), features the majority
of polynyas seen in observations (Fig. 5, A5). It is remarkable that all the coastal polynyas found in the observational product
find their counterpart in the model. Even the shapes and occurrence probability, for example of the Ross Sea polynya or the
polynyas along the East Antarctic coast, are similar.

In agreement with observations, the models show small coastal polynyas during winter that grow in size and in time, merging
with other polynyas in spring/summer (Nov-Jan, see Fig. 5b or supp. Fig. A5). The coastal polynyas are generally located
towards the west of geographical features along the continent outlines (Fig. 3, A3, A4) where the polar easterlies prevail
(Barber and Massom, 2007), except for polynyas at the southern end of the western Antarctic Peninsula, which is under the
influence of the westerlies and the ACC and thus produces polynyas that are located towards the east of the land. Our analysis
includes eight model families (e.g. ACCESS, BCC, CESM2...) that participate in CMIP6 with different model versions (Table
1). Seven of these families show polynyas at very similar locations for all members inside one family (Fig. 3, A3, A4); the only
exception is the ACCESS models. This indicates that the position of polynyas is mostly determined by the model properties and





**Figure 3.** Spatial distribution of polynyas in the CMIP6 models, computed from the sea ice thickness output for September. The red-to-yellow colours indicate the number of years where polynyas occurred within the 165 years of the historical model run for each grid cell. The blue colors show the average sea ice concentration (in %).

less by coincidence of sporadic polynya occurrence or long term variability. Comparison of the results from sea ice thickness (Fig. 3) and sea ice concentration (Fig. A4, A3) shows similar locations, but we found on average almost twice the polynya area by the sea ice thickness threshold method (Table 2).





Half of the 27 models show OWPs in the Ross or the Weddell Seas. These polynyas are most common close to Maud Rise
in the Weddell Sea (Fig. 3, A3, A4). Only 10 models show recurring OWPs in the Weddell Sea: ACCESS-ESM1-5, BCC-
(CM2-MR/ESM1), CAMS-CSM1-0, EC-Earth3(-Veg), GFDL-ESM4, HadGEM3-GC31-LL, and MPI-ESM1-2-(LR/HR). Of
the 23 (28) models that provided sea ice thickness (concentration) output, 6 (8) show, on average, larger OWP than coastal
polynya areas, while the remaining 17 (20) models show more coastal polynyas (Fig. 6). Five (fifteen) models overestimate the
total polynya area compared to sea ice thickness (concentration) observations. Nine models (ACCESS-CM2, CAMS-CSM1.0,
CNRM-CM6, CNRM-CSM2, GFDL-ESM4, all MPIs, UKESM) underestimate the polynya area when derived from the sea
ice thickness (12 cm threshold) but at the same time overestimate it when derived from sea ice concentration (30% threshold).
Assuming the observational sea ice concentration product is accurate, this indicates these nine models show too low sea ice
concentrations within regions covered by thin sea ice. Overall, the multi-model-spread in the polynya area is large, no matter
if it is derived from monthly sea ice thickness, monthly concentration or daily concentration ranging from almost no polynyas
(SAM0-UNICON) up to 214 930 km$^2$, 64 470 km$^2$, or 80 220 km$^2$ respectively for EC-Earth3-Veg. The multi-model mean
is almost twice as high as the observed area (21 880 km$^2$ compared to 10 640 km$^2$ for monthly concentration, 38 560 km$^2$ to
12 050 km$^2$ for daily concentration and 58 680 km$^2$ to 73 060 for monthly sea ice thickness). This is driven primarily by five
models; MPI-ESM1-2-HR, ACCESS-CM2, CNRM-ESM2, and the EC-Earth3s all overestimate the polynya area by a factor
of five.

Most models (15 when using thickness, 21 concentration) show a larger fraction of coastal polynya than OWP area, while
in the rest the OWPs prevail as a larger fraction (Fig. 6). Surprisingly the model ACCESS-CM2 shows only coastal polynyas
while ACCESS-ESM1-5 shows an overabundance of OWPs. These two models share similar versions of the ACCESS-OM2
ocean model component and CICE sea ice model component (Table 1). We further investigate the cause for this difference and
its effect on the water stratification in section 5.3.

**4.2 Polynya frequency and seasonality**

In Section 4.1, we discussed the average polynya areas over 35 years. A closer look at individual yearly values from the full
historical run reveals that the polynya areas vary from year to year (Figures 7); for the coastal polynyas the ratio between the
year with the largest polynya area to that with the smallest one is at least 2.5 (lowest for MRI-ESM2, highest for CESM2-FV2,
HadGEM3-GC31-LL, MPI-ESM-1-2-HAM, SAM0-UNICON, which have one ore more years without any coastal polynyas).
Models with large coastal polynyas usually have little or no OWPs, while models with large OWPs have few coastal polynyas
(Figure 7). This is most distinct for comparison between the ACCESS models, but can also bee seen in comparison of e.g. MPI
to CESM2 models or CAMS-CSM1.0 to MRI-ESM2. The models with a larger fraction of coastal polynyas are ACCESS-
CM2, all CESM2s, CNRMs, EC-Earth3s, IPSL-CM6A-LR, MRI-ESM2.0 and UKESM1-0-LL, for the rest the OWPs are
dominating. Heuzé (2021) found a negative bottom density bias in the deep ocean for the latter, they form the Antarctic Bottom
Water frequently via deep convection instead of shelf processes. We believe that the deep convection is the reason for the OWPs.
Models without the negative bottom density bias show predominantly coastal polynyas (CESM2s, CanESM5, CNRM-CM6.1,
CanESM5, IPSL-CM6A, SAM0-UNICON, UKESM1-0-LL).





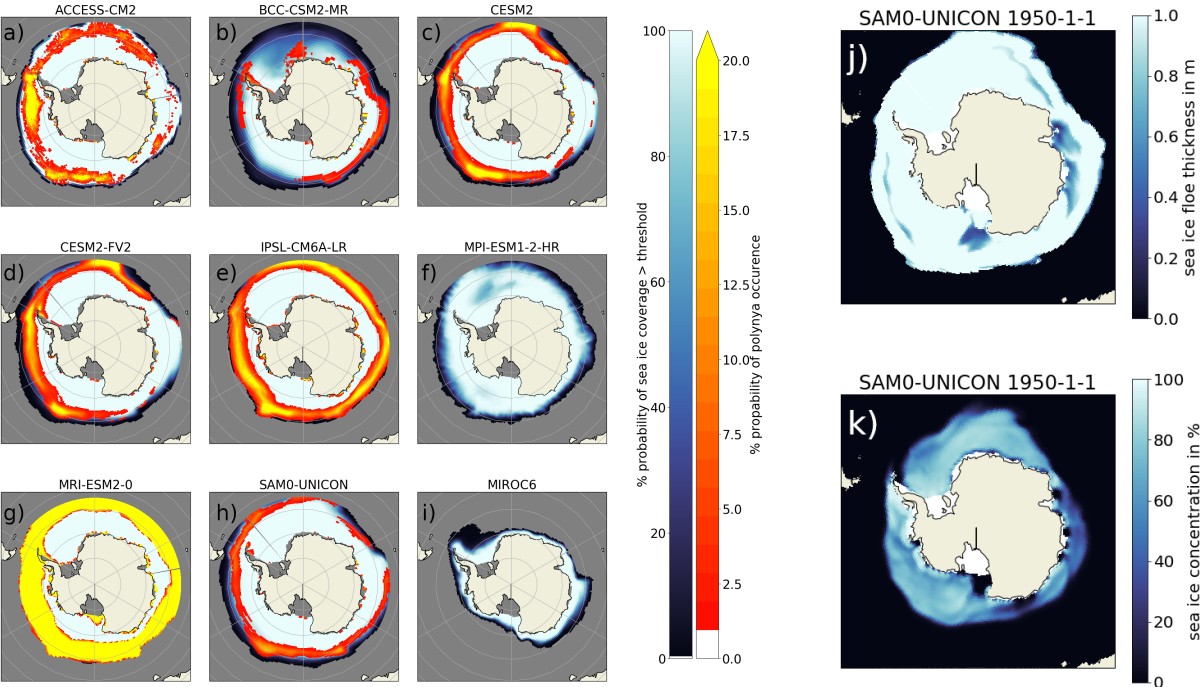

**Figure 4.** **(a)-(i)**: 'Polynya areas' computed with a sea ice thickness threshold method from the ice floe thickness output data. **(j)-(k)**: SAM0-UNICON sea ice floe thickness and sea ice concentration for one time step.

The distribution of the OWPs has one fundamental difference to the one for the coastal polynyas. All models except for the BCCs show years without any or with negligible OWP activity (<1000 km$^3$). The OWP activity we have found in Section

4.1 for the EC-Earth3, GFDL and MPI models is mostly caused by some isolated, large scale OWP events while the majority of years have little to no OWPs. OWPs in the models are not randomly distributed in time but instead regularly re-open for several consecutive years, as seen in the MPI, ACCESS, BCC, CAMS-CSM1 and EC-Earth models (Fig. 8, A7, A8). This is interesting since the sea ice in the Weddell Sea melts almost completely every summer. Thus polynya formation must either be facilitated by an external forcing that persists over several years (Campbell et al., 2019,  suggest positive SAM fluctuations),

or the vertical mixing in open water polynyas acts as a preconditioning for following years (Martinson et al., 1981). We will discuss this further in Section 4.3. While ACCESS-CM2 and BCC-(ESM1/CSM2) show OWPs almost every year, MPI-ESM1.2-(LR/HR), EC-Earth3 and CAMS-CSM1 show episodes of high OWP activity separated by multiple decades of little



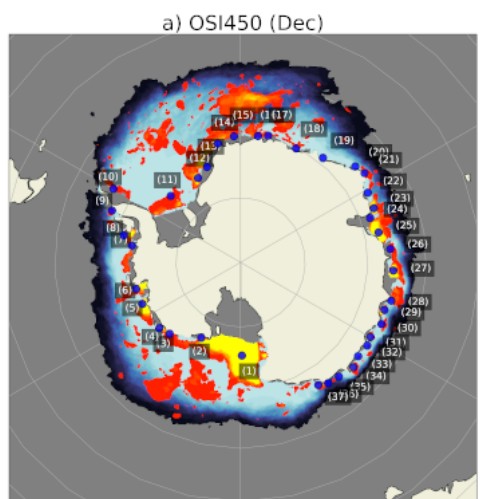
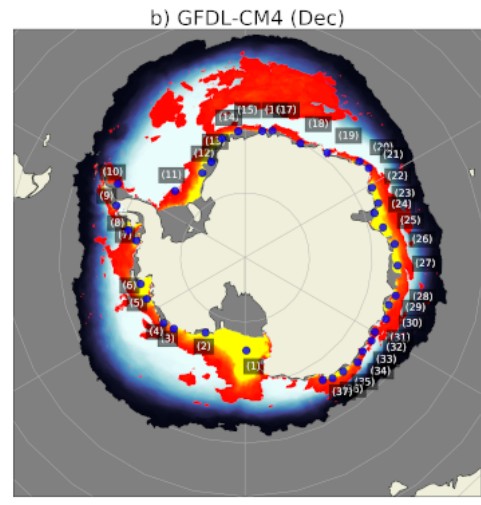
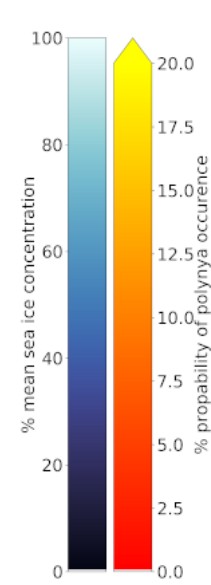

**Figure 5.** Frequency and location of polynya occurrences in December (colours) with the positions of coastal polynyas (blue dots, locations from Arrigo and Van Dijken, 2003) for the common time interval of **(a)** the observed satellite sea ice (OSI450) and **(b)** the GFDL-CM4 CMIP6 model with the highest spatial resolution (25x25 km).

to no OWPs, as is the case for the real-world Weddell Polynya (5 out of 50 years of observation, bottom of Fig. 8, A7, A8 and Carsey, 1980; Campbell et al., 2019).

For the majority of models, the coastal polynyas appear randomly distributed over the years (Fig. 8, A7, A8). In agreement with observations, the models show the largest coastal polynyas between November and January (Fig. 9b,c). However, the models underestimate the coastal polynya area in November and December followed by an equally strong overestimation for January and February (Fig. 9b). The season with the highest OWP activity in the CMIP6 models is between August and November (Fig. 9c,f), while the the sea ice observations show a local maximum in polynya occurrence during ice formation

in June and an absolute maximum during ice melt in November/December. The seasonality of the OWPs does not agree well between CMIP6 and observations.

### 4.3    Vertical ocean stratification in OWPs

OWPs are strongly influenced by upwelled sensible heat and they are a source of bottom water production, both in observations (Cheon and Gordon, 2019) and in CMIP models (Heuzé et al., 2013), hence we expect to see a clear difference in the water

stratification inside an open water polynya compared to the stratification under sea ice. To evaluate the effect of OWPs on the water column, we separate the vertical profiles of the CMIP6 models in the Weddell Sea (between 65°W and 30°E) during the presence or absence of an OWP. Again, we use only the Weddell Sea because it shows the most OWP activity in CMIP6 models (Fig. 3, A3, A4) and observations (Campbell et al., 2019).



To evaluate the effect of OWPs on vertical stratification, we concentrate on the models that form most OWPs (see Table 2) and show only the top three models (MPI-ESM1.2-HR, ACCESS-ESM1.5, BCC-ESM1) in Fig. 10. For comparison, we present the observed hydrographic data of a SOCCOM profiling float (Johnson et al., 2018), which surfaced in the Maud Rise Polynya in winter 2017.

In the three CMIP6 models, we find major differences in the vertical stratification between the profiles under sea ice and inside OWPs (Fig. 10). In polynyas, deep mixing is resulting in a very weakly stratified water column down to 4000 m, and in particular the salinity difference between mixed layer and Circumpolar Deep Water is reduced by up to 80% (not shown). This is mainly due to an increased salinity in the upper 1000 m. Close inspection reveals that there is also a reduction in salinity in the Circumpolar Deep Water. Especially for the models MPI-ESM1.2-HR and ACCESS-ESM1.5 we also see a shoaling of the temperature maximum from 824 m to 645 m and 1560 m to 1150 m depth, respectively, and increased temperature from 0.1°C to 0.2°C for both models in the upper 500 m of the averaged profile. In the observational profiles (Fig. 10) the temperature maximum can be found at a depth of about 200 m. ACCESS-ESM1.5 and BCC-ESM1 underestimate the temperature maximum by 1.5°C and 1°C respectively, and show it to occur too deep: $1148 \pm 132$ m and $1366 \pm 282$ m, respectively under the sea ice; and $1560 \pm 438$ m and $1205 \pm 389$ m, respectively in OWPs.

In agreement with Cheon and Gordon (2019), we also find that the temperature profiles under the sea ice are more uniform than the profiles within the polynya (Fig. 10). The coldest profiles within polynyas are in fact the least stratified ones. For the model MPI-ESM1.2-HR for example, the under sea ice temperature varies about 0.5°C from the mean value, but varies by more than 1°C inside OWPs. A possible explanation is that there is available heat upon the opening of polynyas (e.g. due to Ekman pumping, warm profiles), but then rapid cooling of the open surface water in the winter occurs. If the stratification was already weak or the density at the surface is increased by brine rejection of newly formed sea ice, deep convection may commence (weakly stratified cold profiles, Fig. 10).

Active mixing processes reach different depths in the CMIP models (Fig. 10). MPI-ESM1.2-HR contains some remaining stratification in all profiles, ACCESS-ESM1.5 shows evidence for mixing all the way to the bottom in many profiles, while BCC-ESM1 conserves some stratification close to the bottom. In comparison to Argo float measurements, the models show deeper reaching mixing with a portion of the profiles showing no substantial stratification to the bottom (5000 m). However, not all profiles from inside an OWP are homogenised by mixing. There are some profiles for each of these models that show a warmer, shallower temperature maximum compared to the under ice case. This results in a shallower temperature maximum on average (Fig. 10). In comparison to Argo float data, the vertical mixing in the models seems overestimated, which was expected because we chose the models with the highest OWP activity. Moreover, the float data features only a few profiles from the outer edge of one Maud Rise Polynya event (Campbell et al., 2019). A discussion of the stratification changes from the 1970s Weddell Sea Polynya can be found in Cheon and Gordon (2019) and shows similar mixing as the CMIP6 models.

Re-arranging the profiles into Temperature-Salinity diagrams (right of Fig. 10), the core of the comparatively warm Circumpolar Deep Water (CDW), at approximately 1000 m depth, is easily visible from the temperature maximum. Under sea ice, the water masses on either side of the CDW are clearly separated: the surface layer is cold and fresh, while the deep water is saltier and warmer than the surface. The pycnocline between these layers is at a depth of 100 to 200 m, except for the BCC models,



**Table 2.** Mean yearly areas of coastal polynyas (pa_co), open water polynyas (pa_op), their sum (pa_tot) and the maximum sea ice area (sa_tot) in $10^3$km$^2$ derived from monthly/daily sea ice concentration and monthly sea ice thickness for September for the complete Southern Ocean, averaged over 1979-2015. Observational sea ice thickness marked with a (*) is available over a 10 year period only. The four models matching the observations closest are printed **bold**, *italic models* have a r-value less than 0.8 (Fig. A1, A2) and thus a relatively weak consistency over the three used sea ice variables.

| Model | concentration monthly | | | | concentration daily | | | | thickness monthly | | | |
|---|---|---|---|---|---|---|---|---|---|---|---|---|
| | pa_co | pa_op | pa_tot | sa_tot | pa_co | pa_op | pa_tot | sa_tot | pa_co | pa_op | pa_tot | sa_tot |
| OBSERVATIONS | 8.53 | 2.11 | 10.64 | 18,303.27 | 8.41 | 3.64 | 12.05 | 18,500.67 | 59.10* | 13.96* | 73.06* | 17,190.85* |
| ACCESS-CM2 | 47.53 | **1.12** | 48.65 | 15,790.74 | 83.23 | **2.03** | 85.25 | 15,838.08 | **54.58** | 1.31 | 55.89 | 15,900.04 |
| ACCESS-ESM1-5 | 0.91 | 58.78 | 59.69 | **16,968.50** | – | – | – | – | 8.86 | 53.67 | **62.53** | **17,178.95** |
| *BCC-CSM2-MR* | 9.04 | 3.28 | **12.32** | 14,232.37 | 18.06 | **2.51** | **20.57** | 14,333.47 | 2.80 | 148.20 | 151.00 | 20,108.89 |
| BCC-ESM1 | 14.53 | 3.54 | 18.07 | 15,095.62 | 16.09 | 25.39 | 41.48 | 15,442.34 | 11.46 | 71.85 | **83.31** | **17,658.59** |
| CAMS-CSM1-0 | 10.87 | 3.90 | **14.76** | 16,055.20 | – | – | – | – | 6.36 | 24.41 | 30.77 | 20,274.28 |
| CESM2 | 2.94 | 0.00 | 2.94 | 16,249.46 | 3.22 | 0.05 | 3.26 | 16,309.43 | 31.58 | 0.02 | 31.61 | 16,086.87 |
| CESM2-FV2 | 5.81 | 0.04 | 5.85 | 16,508.26 | 5.76 | 0.25 | **6.01** | 16,581.80 | 14.19 | 0.00 | 14.19 | **17,000.16** |
| CESM2-WACCM | 3.10 | 0.00 | 3.10 | 16,860.61 | 3.71 | 0.04 | 3.75 | **16,926.73** | 29.32 | 0.00 | 29.32 | 16,743.45 |
| CESM2-WACCM-FV2 | 4.97 | 0.00 | 4.97 | 16,355.55 | 3.90 | 0.03 | 3.94 | 16,445.10 | 35.27 | 0.02 | 35.29 | 16,229.04 |
| CNRM-CM6-1 | 20.08 | 0.19 | 20.28 | **18,644.08** | 28.90 | 0.74 | 29.64 | **18,717.85** | **55.09** | 1.39 | 56.47 | 15,713.60 |
| CNRM-ESM2-1 | 48.15 | 0.34 | 48.49 | 16,118.08 | 60.25 | 0.81 | 61.06 | 16,203.09 | 43.56 | 0.49 | 44.05 | 14,263.37 |
| CanESM5 | 24.62 | 11.64 | 36.25 | 19,169.02 | 56.44 | 14.22 | 70.66 | **19,130.14** | – | – | – | – |
| EC-Earth3 | 55.27 | 4.89 | 60.16 | 11,658.74 | 70.15 | **6.26** | 76.41 | 11,618.18 | 200.10 | 3.07 | 203.17 | 10,389.41 |
| EC-Earth3-Veg | 52.89 | 11.59 | 64.47 | 11,724.48 | 67.22 | 13.00 | 80.22 | 11,670.90 | 207.70 | **7.23** | 214.93 | 10,322.34 |
| GFDL-CM4 | 2.49 | **2.64** | 5.13 | 20,196.86 | – | – | – | – | 17.59 | 2.35 | 19.94 | 19,405.82 |
| GFDL-ESM4 | 4.69 | 14.45 | 19.14 | **18,360.44** | – | – | – | – | 23.47 | **12.55** | **36.03** | **17,428.99** |
| HadGEM3-GC31-LL | 3.30 | 5.74 | **9.05** | 15,231.28 | – | – | – | – | 2.77 | **4.61** | 7.39 | 15,426.68 |
| *IPSL-CM6A-LR* | 4.26 | 0.06 | 4.32 | 20,516.08 | **12.98** | 0.38 | **13.36** | 20,443.25 | 97.43 | 0.38 | **97.81** | 19,276.91 |
| MIROC-ES2L | 14.70 | 0.00 | **14.70** | 3,899.66 | – | – | – | – | – | – | – | – |
| MIROC6 | **6.06** | 0.00 | 6.06 | 3,963.83 | – | – | – | – | – | – | – | – |
| MPI-ESM-1-2-HAM | 3.65 | 12.22 | 15.87 | 9,098.21 | **14.15** | 15.31 | 29.46 | 9,355.54 | 1.93 | **13.22** | 15.15 | 9,585.31 |
| MPI-ESM1-2-HR | **7.88** | 44.32 | 52.21 | 15,896.38 | 31.57 | 58.50 | 90.07 | 16,174.62 | 8.59 | 29.14 | 37.72 | 15,117.01 |
| MPI-ESM1-2-LR | 3.22 | 33.17 | 36.40 | 12,615.25 | 14.51 | 43.74 | 58.25 | 12,951.94 | 1.91 | 38.06 | 39.97 | 13,233.24 |
| *MRI-ESM2-0* | 0.15 | 0.00 | 0.15 | 20,825.05 | 2.37 | 0.00 | 2.37 | 20,894.15 | **63.37** | 0.00 | **63.37** | 18,307.11 |
| NorCPM1 | 4.29 | 0.08 | 4.37 | 20,637.46 | – | – | – | – | – | – | – | – |
| SAM0-UNICON | 5.96 | 0.00 | 5.96 | **19,441.75** | 17.88 | 0.08 | **17.96** | **19,513.31** | 6.50 | 0.00 | 6.50 | 19,474.05 |
| UKESM1-0-LL | 16.55 | 0.90 | 17.46 | 16,718.76 | – | – | – | – | 12.39 | 0.91 | 13.30 | 16,907.96 |





which show very weak stratification and no clear water mass layers can be identified. Inside the polynyas, the differentiation of
the water masses is almost gone (Fig. 10). The models with the highest OWP activity (ACCESS-ESM1.5, BCC and MPI) have
low $N^2$ values even under sea ice, where they hardly exceed 0.5 $10^{-5} s^{-2}$ (Fig 10). The deep mixing is transporting heat and
salt into the surface layer, reducing its density difference compared to deeper layers. This weakened stratification can remain
in the water for a significant period of time, preconditioning the region for new polynyas to form in following years (Martinson
et al., 1981).

## 5 Discussion

In the previous sections, we looked at the representation of polynyas in CMIP6 models and their effect on the local water
stratification. We found large variations among models, with the majority underestimating OWP area but a few largely over-
estimating it. The hot spot for OWPs in CMIP6 and observations is the Weddell Sea. Now we want to turn our attention to
finding the causes for the difference in polynya activity in the CMIP6 models. We start with a comparison of Climate models
versus Earth System models and then discuss whether the model's winds and Antarctic Circumpolar Currents can explain its
OWP representation. Then we will examine the possible reasons for the overestimation of OWPs in many models and give an
outlook about how we expect these biases to change in the near future.

### 5.1 Polynyas in CM versus ESM versions

The ACCESS, BCC and GFDL families each have two different model configurations for CMIP6, a climate model (CM)
version and an Earth system model (ESM) version. In addition to climate physics, ESMs include physical, chemical and
biological processes and interactions and can therefore lead to more realistic climate predictions. Dong et al. (2020) found that
the ESM-versions have less climate sensitivity compared to the CM-versions. We find that the ESM versions show more OWP
activity (Fig. 6). This is consistent with the results of De Lavergne et al. (2014), in which OWPs in the Weddell Sea eventually
stop at the end of the extended CMIP5 climate change runs, as the CM models show a stronger warming response than the
ESM-versions Dong et al. (2020).

CM models can usually run at a higher resolution, due to their reduced complexity in comparison to ESMs. In our case,
GFDL-CM4 has a higher oceanic resolution than GFDL-ESM4 and BCC-CSM2 has a higher atmospheric resolution than
BCC-ESM1 (Table 1). A high spatial resolution can improve the representation of katabatic winds and coastlines, which might
be why the above mentioned CMs have a larger fraction of coastal polynyas than their ESM counterparts (Fig. 6). The CM
model versions do not only show larger polynya areas (Fig. 6 and Fig. 3) but they also show more frequent occurrences of OWPs
(Fig. 8). The same trend can be seen between the MPI models, which are alike except for their difference in resolution. The
strongest difference in the positions and type of polynyas is found between the ACCESS-CM2 and ACCESS-ESM1.5 models,
which show primarily coastal or open water polynyas respectively (Fig. 3, A3, A4). This difference is further discussed in
section 5.3.





## 5.2 Does strong wind forcing and ACC transport facilitate OWPs?

Open water polynya activity is commonly attributed to the Southern Annular Mode, the strength of the Southern Hemisphere winds and the transport of the ACC (e.g. Swart et al., 2018; Cheon et al., 2014; Campbell et al., 2019; Behrens et al., 2016). We combine our polynya characteristics from section 4 with the statistical evaluation of Southern Ocean properties from Beadling et al. (2020) to determine potential across-model relationships between Southern Ocean properties and polynya activity. Since Beadling et al. (2020) presents data from 1986-2005, we restrict our analysis to the same period and the 23 models we have in common. The CMIP6 across-model correlation between the total wind stress forcing of the westerlies (Beadling et al., 2020) with OWP activity (Table 2) is 0.4: that is, the models with the strongest Southern Hemisphere Westerlies have the highest OWP activity, while the models with the weakest westerlies show no OWP activity. We found this correlation to be the same for polynya detection from sea ice concentration and sea ice thickness

All observed Maud Rise polynyas were preceded by a positive Southern Annular Mode (Campbell et al., 2019). During a positive phase, the westerly wind belt intensifies and contracts towards Antarctica. This process leads to an increased cyclonic wind stress curl and intensifies the Weddell Gyre (Cheon et al., 2014). For the CMIP6 models we found that the maximum strength of the zonally averaged wind stress curl correlates with its latitude (correlation coefficient of -0.82, computed again using values from Beadling et al., 2020, , see Fig. A6.). That means that, unsurprisingly, the CMIP6 models have different average SAM index values. Consequently, we expected an increased OWP activity in the Weddell Sea for the models with the highest wind stress curl maximum. This would consolidate the correlation between SAM and polynya activity. In contrast to Campbell et al. (2019) we find no significant correlation between the wind stress curl maximum and the average OWP area. CMIP6 models temporarily or constantly in a phase with a high SAM index do not show significantly increased OWP activity.

As shown by Hirabara et al. (2012), the ACC transport is positively correlated with OWP activity in observations. Models with an overestimation of OWPs also have too much deep convection (Heuzé, 2021, and Figure 10), which increases the meridional density gradient and thus enhances the ACC transport (Hirabara et al., 2012; Beadling et al., 2020). The CMIP6 models that are featuring an ACC in agreement with observational values are ACCESS-CM2, ACCESS-ESM1-5, BCC-ESM1, CanESM5, GFDL-ESM4, IPSL-CM6A-LR, MIROC6, MPI-ESM-1-2-HAM, MPI-ESM1-2-LR and UKESM1-0-LL (Beadling et al., 2020). Nine of these ten models exhibit open ocean convection with large OWPs in the Southern Ocean (Heuzé, 2021, see the models marked with a '*' in Table A1). The exception is MIROC6, which does not form sufficient sea ice for OWPs to occur but still shows extensive deep convection (Tatebe et al., 2019; Heuzé, 2021). In summary, we find an across-model relationship between the strength of the ACC and OWP activity; models with an overestimation of OWPs are more likely to simulate a realistic ACC transport. An explanation for this bias could be a general warm bias of the Circumpolar Deep Water (Sallée et al., 2013, found for CMIP5 models) that causes OWPs, rather than under-sea-ice deep convection (Dufour et al., 2017).

We have shown that overabundant OWPs are a common issue in models (sections 4.1 and 4.2); that is a problem because it affects the whole water column (section 4.3). We have now found relationships with the models' wider representation of the



Southern Ocean, but these relationships have some exceptions. In a case study, we will now investigate the specific case of two seemingly similar models that have extremely different OWP representations, ACCESS-CM2 and ACCESS-ESM1.5.

### 5.3 Overestimation of OWPs in models and future perspectives

We found that most models overestimate polynya activity in comparison to the observations (Table 2). We believe this to be general known in the community given that within model documentation, there are many descriptions of approaches to reduce deep convection and the presence of OWPs in the Southern Ocean (listed and highlighted with a '*' in supplementary Table A1). We first describe approaches to reduce OWPs and conclude with future perspectives.

In the previous section, we discussed the correlation of the wind stress curl and ACC transport to polynya activity; we could confirm this in a multi model correlation for the ACC. However, Beadling et al. (2020) found that 30 of 35 CMIP6 models underestimated the ACC, 34 of 38 showed their wind stress curl minimum not sufficiently south and 33 of 38 underestimated the wind stress curl maximum. All these parameters are positively correlated with OWP activity (Campbell et al., 2019). If future model generations are approaching more realistic values for ACC strength, wind stress curl and SAM, we expect that the problem of open water convection with the formation of large OWP will become evident for even more models.

One of the development foci for the GFDL models was to reduce Southern Ocean polynyas (Held et al., 2019). For the GFDL-CM2 models, the formation of super polynyas in the Southern Ocean was addressed by increasing the near-infrared albedo of glaciers and snow covered ice caps in order to increase coastal freshwater inflow. This is reported to delay the formation of super polynyas in the model, but not to prevent them completely (Held et al., 2019).

The CESM2 models feature an overflow parameterisation (Briegleb et al., 2010), which can transport dense bottom water down the shelf and should help with more realistic bottom water formation. This can lead to a more stable stratification, but Heuzé (2015) found that it also caused an overestimation of coastal polynyas in their CMIP5 model version. In our study, we find that the CMIP6 CESM2 models show coastal polynyas, but never OWPs. The observed polynya area is underestimated by at least 30% by all analyzed CESM2 models.

We found that both ACCESS models show more than double the polynya area of the observational data (Fig. 6, Fig. 9). Even though the ACCESS-ESM1.5 and the ACCESS-CM2 models share most model components, the former shows prevalent OWPs while the latter has mostly coastal polynyas. In the assessment of ACCESS-ESM1.5, Bi et al. (2013) found that OWPs in the Weddell Sea form too often due to deep convection. An iceberg discharge scheme was hence introduced into the model ACCESS-CM2, which freshens and cools the upper ocean (Siobhan O'Farrell, personal communication, June 2020). Freshwater is transported further out into the open ocean instead of entering the ocean directly at the coast. This change effectively suppresses the OWP activity in ACCESS-CM2 (Fig. 3). The freshwater transport of icebergs somewhat suppresses too frequent open ocean deep convection events (Heuzé, 2021), but ACCESS-CM2 forms too large coastal polynyas instead, much more frequent than ACCESS-ESM1.5. The extent of these coastal polynyas is unrealistically high and the problem of overall too large polynya areas remains. The suppressed OWP activity due to freshwater discharge from icebergs affects the whole water column in the Weddell Sea (Fig. 11). In ACCESS-CM2, the Weddell Deep Water is 0.7°C warmer and the maximum temperature of the CDW is 1.1°C warmer than in ACCESS-ESM1.5.



Here we discussed some example of modellers' approaches to reduce deep convection and OWPs in the Southern Ocean. In the model documentations (listed in supplementary Table A1), we found more examples with the same aim. On the other hand, several models have found a way to prevent this issues. The CESM2 models have an overflow parameterisation (Briegleb et al., 2010) and do not show any OWPs. This parameterisation can provide an effective way for realistic bottom water formation, until higher model resolutions and further model improvements (e.g. better vertical discretisation schemes) allow for actual bottom water formation in coastal polynyas. Also, isopycnal coordinates are beneficial for the representation of down slope flows, the deep water formation in coastal polynyas. This in turn results in more realistic deep water properties (Heuzé et al., 2013). We found that no model with isopycnal coordinates (Table 1) shows an unrealistic amount of OWPs (Table 2 and Fig. 6). The Modular Ocean Model (MOM) introduced isopycnal coordinates in its latest version MOM6. Since several models are based on MOM, e.g. the ACCESS and the BCC models, we expect an improvement in deep water formation, Southern Ocean deep water properties and OWPs when these models successively upgrade to MOM6.

## 6   Conclusions

In this paper, we evaluated the representation of Southern Ocean open water and coastal polynyas in CMIP6 climate models and their effects on the modelled Weddell Sea. We found coastal polynyas in all 27 analysed models around the Antarctic continent, while OWPs are present in half of the models, either in the Weddell or the Ross Seas. Based on the results from the sea ice thickness, the models ACCESS-ESM1.5, BCC-ESM1 and MRI-ESM2.0 show the best agreement in total polynya area with less than 15% deviation, but ACCESS-ESM1.5 and BCC-ESM1 show a strong bias towards open water polynyas and MRI-ESM2.0 has only coastal polynyas. With total polynya areas from 6.5 x $10^3$ km$^2$ up to 215 x $10^3$ km$^2$, the CMIP6 models show a large model spread. This inconsistent representation of polynyas can be problematic, since polynyas account for large amounts of sea ice and deep water production. We found larger polynyas with the sea ice thickness threshold method in 19 of 23 models and the observational data, four models showed larger total polynya areas with the sea ice concentration threshold method. Using monthly sea ice concentration data instead of daily data leads to an underestimation of the polynya area in all models and the observations.

In section 4.3 we have shown how OWPs significantly reduce the local stratification. We found indications of upwelling as well as deep convection, which eventually change the long term water properties in the Weddell Sea, notably removing the stratification and hence making future polynyas more likely. In section 5.3 we discussed the long term effect of coastal versus open water polynyas in the Weddell Sea using two different ACCESS model versions.

Half of the CMIP6 models show an over representation of OWPs (Table 2), which is a known problem in the field (e.g. Held et al., 2019; Gutjahr et al., 2019; Sellar et al., 2019), likely caused by unrealistic deep convection events in the Southern Ocean (see Table A1 for references). We have described some of the models' strategies to prevent open ocean polynyas and their caveats. CMIP6 models with realistic representation of the ACC transport and Southern Ocean wind fields commonly show unrealistically increased OWP formation in the Weddell Sea (section 5.2). At the same time, future increases of the horizontal





model resolution, improved vertical discretisation schemes, overflow parameterisations and fresh water influxes will likely help
to achieve more realistic representations of polynyas in the Southern Ocean, and in sea ice covered polar regions in general.

*Code availability.*   The code to reproduce the analysis is freely available at https://github.com/MartinMohrmann/Southern-Ocean-polynyas-
in-CMIP6-models.

*Data availability.*   CMIP6 data are freely available via any portal of the Earth System Grid Federation; a list over the different portals to
download the data can be found at https://esgf-node.llnl.gov/projects/cmip6/.

The observational sea ice concentration products OSI450 and OSI-430-b (Lavergne et al, 2019) are provided by the Norwegian and
Danish Meteorological Institutes online: doi: 10.15770/EUM_SAF_OSI_0008. The observational product for thickness of thin sea ice (SIT)
is available via ftp at https://seaice.uni-bremen.de/data/smos. The float data are freely available via the Southern Ocean Carbon and Climate
Observations and Modeling website (SOCCOM).



**Figure 6.** Winter polynya area for the CMIP6 models and the observational products (SMOS thin ice and OSI450). The polynyas are computed from **(a)** monthly sea ice concentration **(b)** daily sea ice concentration and **(c)** monthly sea ice thickness. The solid colors mark the yearly mean polynya area, while the transparent colors show the yearly maximum polynya area. The lower panels show the difference in yearly mean polynya area if derived from monthly/daily sea ice concentration **(d)** and monthly sea ice concentration/thickness **(e)**, for the subset of models that provided both outputs. All data sets where trimmed to the winter period according to Equations 1 and 2 respectively and then the yearly mean from 1979 to and including 2014 for homogenisation in time was formed; the only exception is the observational sea ice thickness, which is not available before 2010 and is presented in a separate panel at its' full length until November 2020 at the side of panels **(c)** and **(e)**. For the mean values in **(a)**, **(b)** and **(c)**, the black bars mark the range in which 50% of the results are located.







**Figure 7.** Mean total winter polynya area as determined from monthly sea ice thickness for the CMIP6 models and the observational product (SMOS thin ice). For the models, results are shown for the full 165 years of the historical scenario, the observational data set includes the years 2010-2020. Each black dot is an average yearly polynya area for one model. The violin shape indicates the distribution of the values, the most values are in the thickest spot of the violins. The white dot indicates the mean value averaged over all available years.



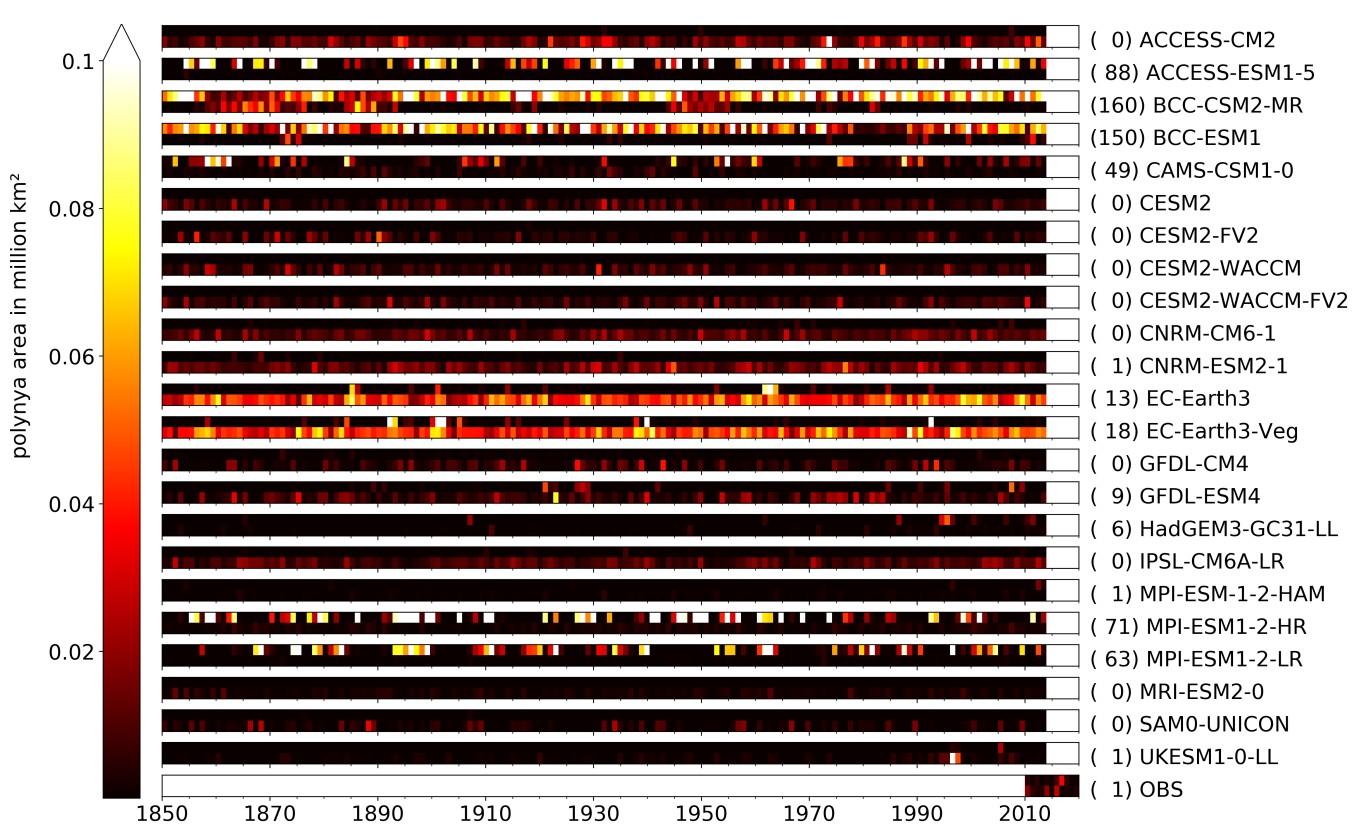

**Figure 8.** Yearly mean polynya area between May and November in historical CMIP6 model run for the Weddell Sea region computed from the sea ice thickness. Each bar represents one CMIP6 model and is divided horizontally into coastal polynya area (upper bar) and OWP area (lower bar). The values next to the model name represent the number of years an open water polynya with an area of more than $10.0 \times 10^3$ km$^2$ was detected.



**Figure 9.** Seasonality of **(a,d)** the sea ice area; **(b,e)** the coastal polynyas area; and **(c,f)** the OWP area, for each CMIP6 model (symbols) and the observation products (plain black lines) in the Weddell Sea. The areas are derived from CMIP6 sea ice thickness and observational SMOS data **(a,b,c)** and from CMIP6 sea ice concentrations and observational OSI450 data **(d,e,f)**. Grey shading indicates a two standard deviation range from the satellite sea ice data. The data was averaged for the time period 1979-2015.

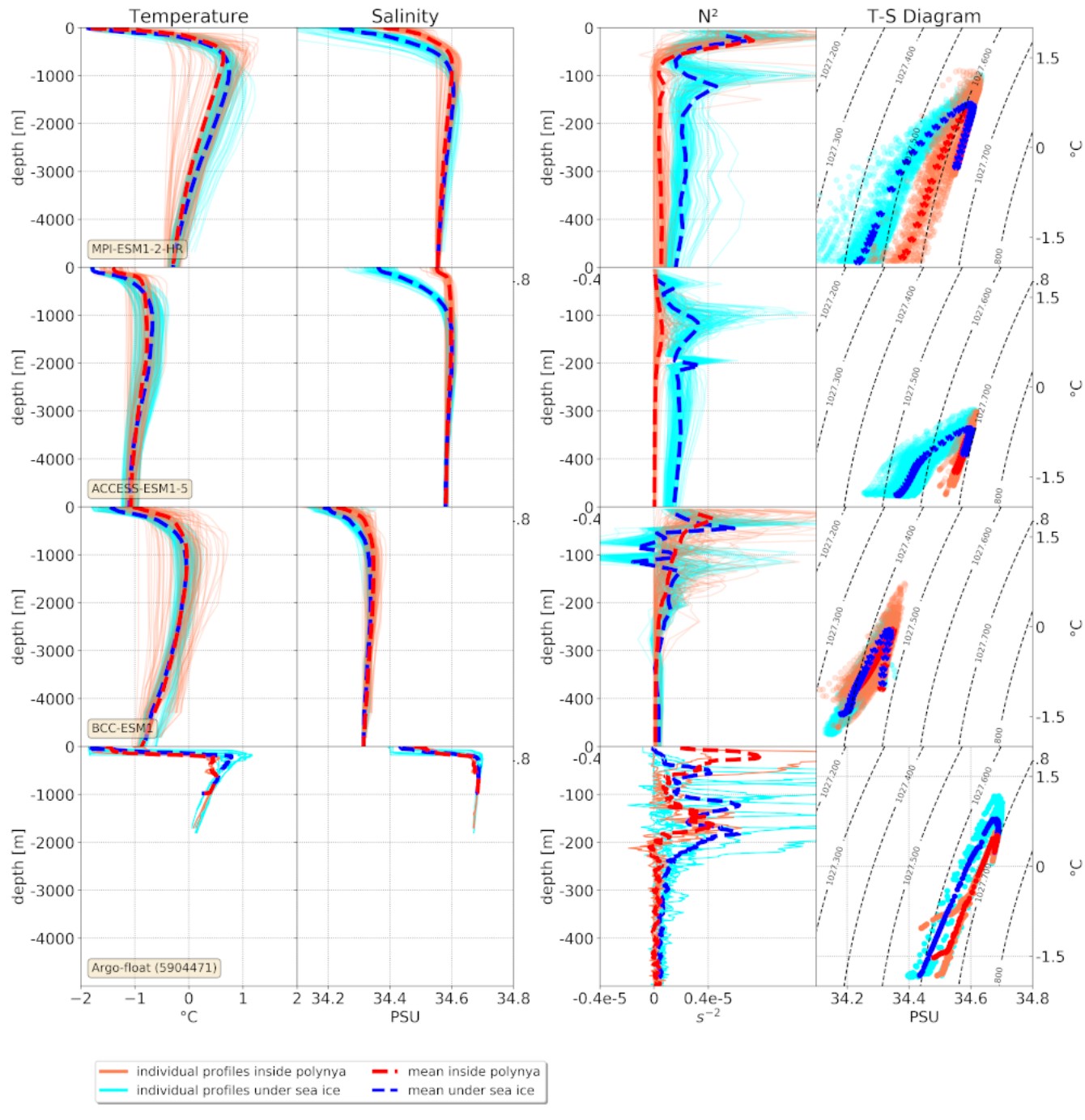

**Figure 10.** Winter profiles occurring within a polynya (red) and under the sea ice (blue, i.e. when no polynya is present) for three representative models and in observations (bottom). From left to right: temperature, salinity, Brunt–Väisälä frequency ($N^2$), and T-S diagram.





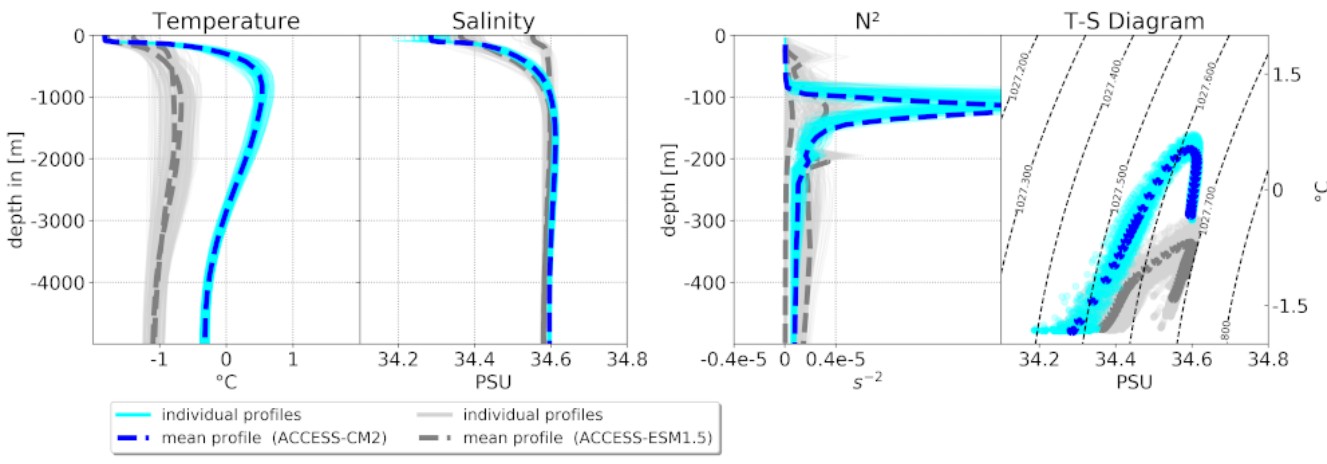

**Figure 11.** Comparison of the winter profiles of the CMIP6 models ACCESS-ESM1.5 (blue) and ACCESS-CM2 (grey). From left to right: temperature, salinity, Brunt–Väisälä frequency $N^2$, and T-S diagram.





**Appendix A: Supplementary Figures and Tables**

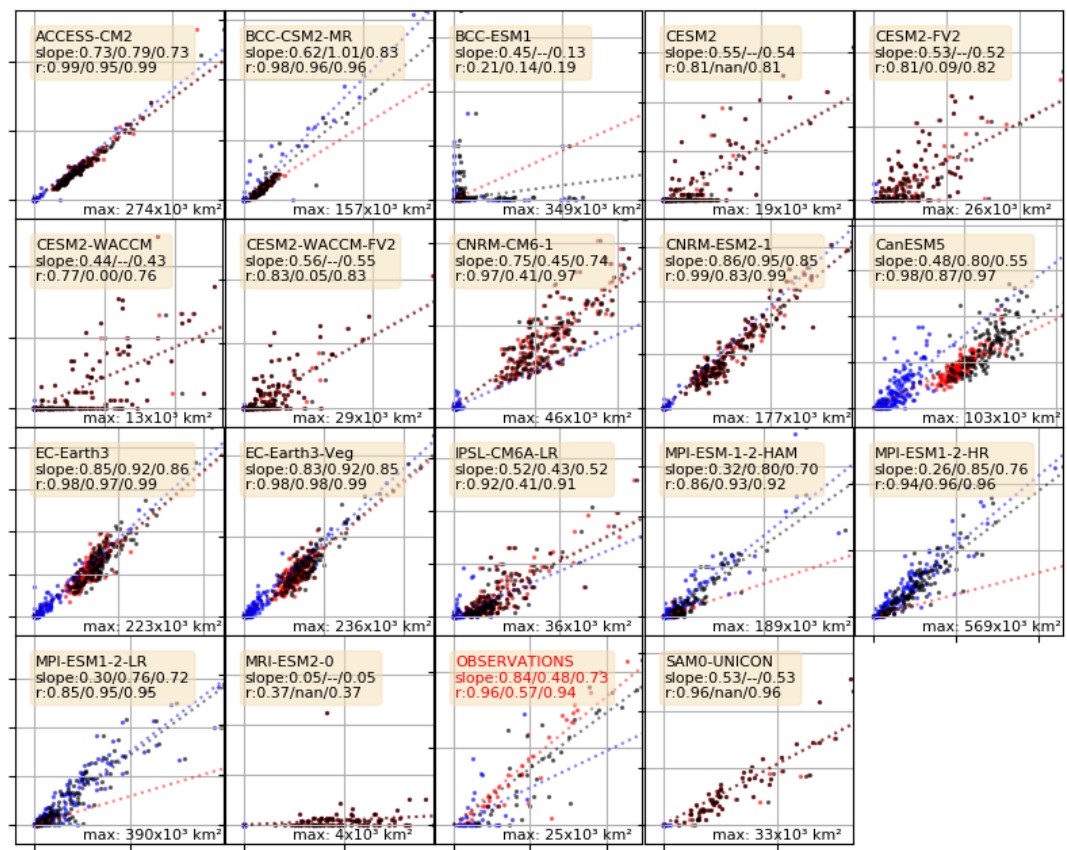

**Figure A1.** Correlation plots for the polynya areas computed from daily (x) vs monthly (y) sea ice concentrations as presented in Table 2. Each dot represents the averaged polynya area of one year; red are the coastal polynyas, blue the open water polynyas and black the sum of both. Where applicable (e.g. polynyas present, p-value < 0.05) the legend includes the slopes of the correlation in the order coastal/open water/combined polynyas. Because the area of polynyas varies by more than one order of magnitude, we normalized the axes of each plot by the highest numerical value for each model and give its value in the lower right corner of each plot.



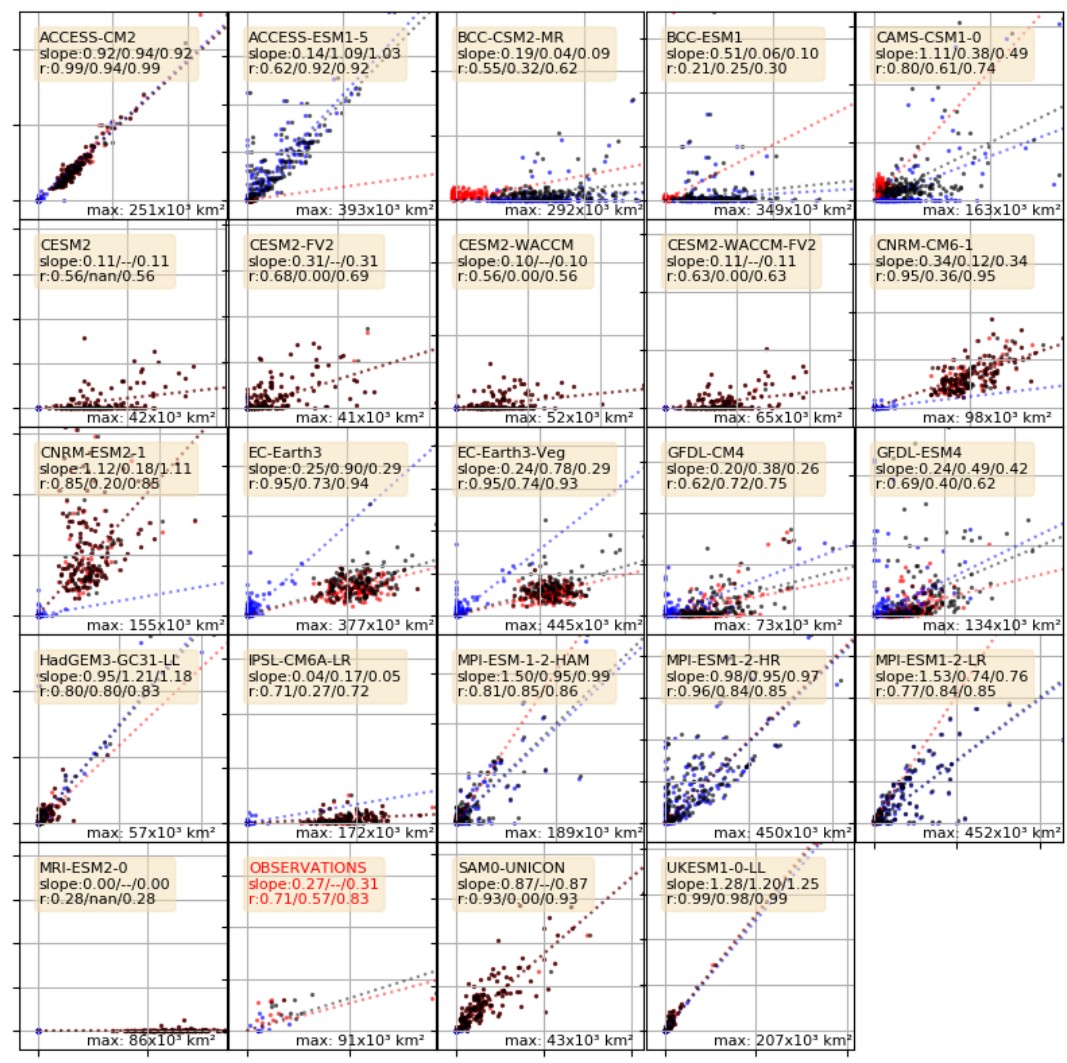

**Figure A2.** Correlation plots for the polynya areas computed from monthly sea ice thickness (x) vs monthly sea ice concentration (y) as presented in Table 2. Each dot represents the averaged polynya area of one year; red are the coastal polynyas, blue the open water polynyas and black the sum of both. Where applicable (e.g. polynyas present, p-value < 0.05) the legend includes the slopes of the correlation in the order coastal/open water/combined polynyas. Because the area of polynyas varies by more than one order of magnitude, we normalized the axes of each plot by the highest numerical value for each model and give its value in the lower right corner of each plot.





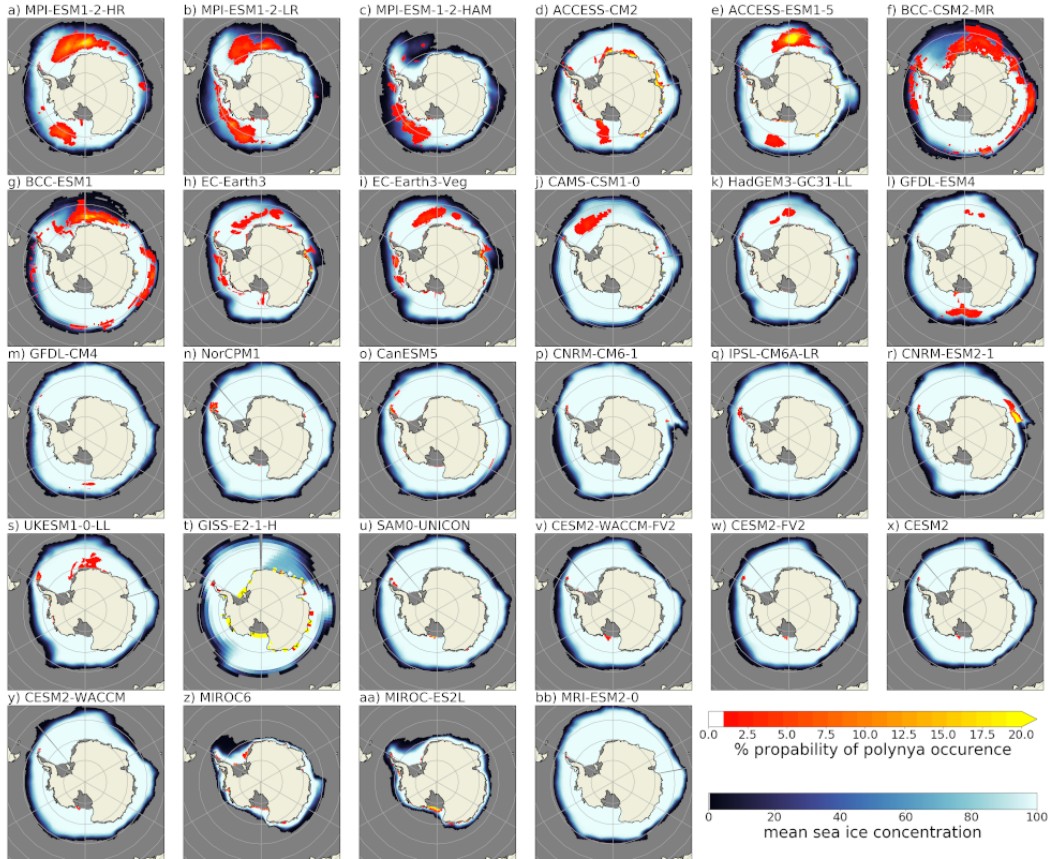

**Figure A3.** Spatial distribution of polynyas in the CMIP6 models, computed from the monthly sea ice concentration output for September. The red-to-yellow colours indicate the number of years where polynyas occurred within the 165 years of the historical model run for each grid cell. The blue colors show the average sea ice concentration (in %).



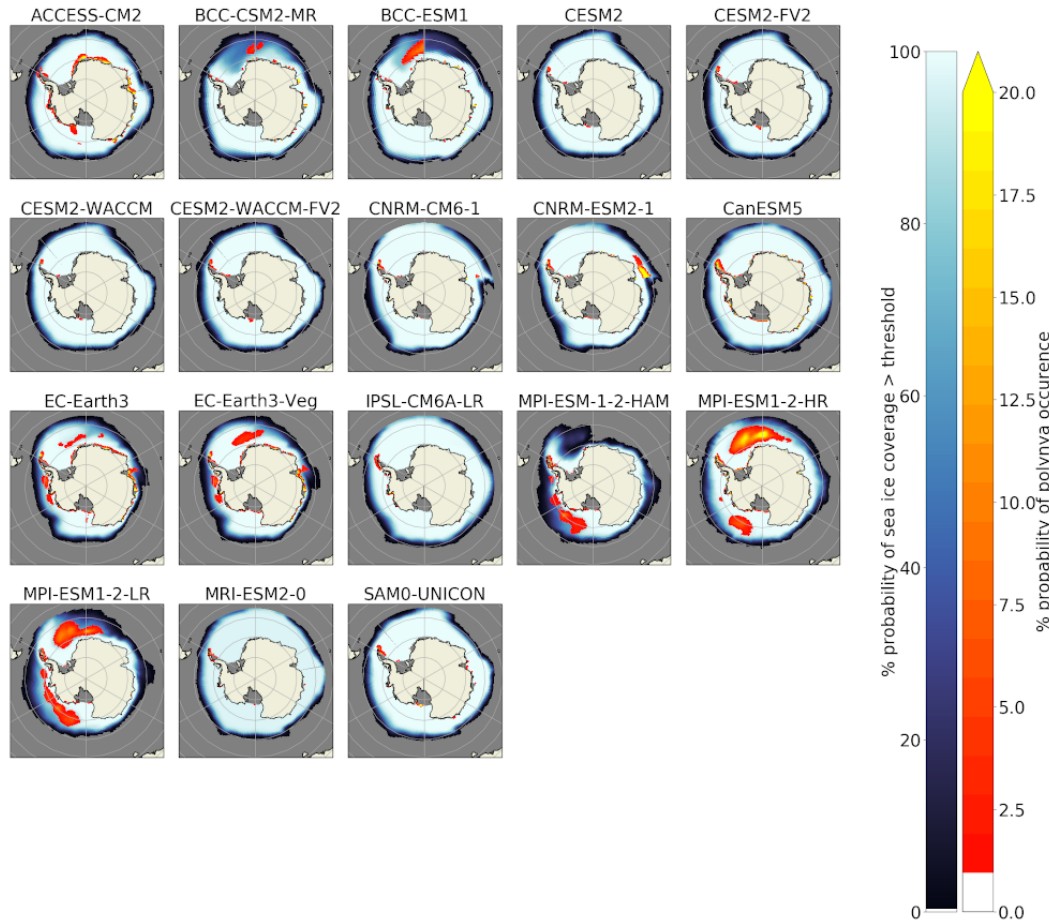

**Figure A4.** Spatial distribution of polynyas in the CMIP6 models, computed from the daily sea ice concentration output for September. The red-to-yellow colours indicate the number of years where polynyas occurred within the 165 years of the historical model run for each grid cell. The blue colors show the average sea ice concentration (in %).





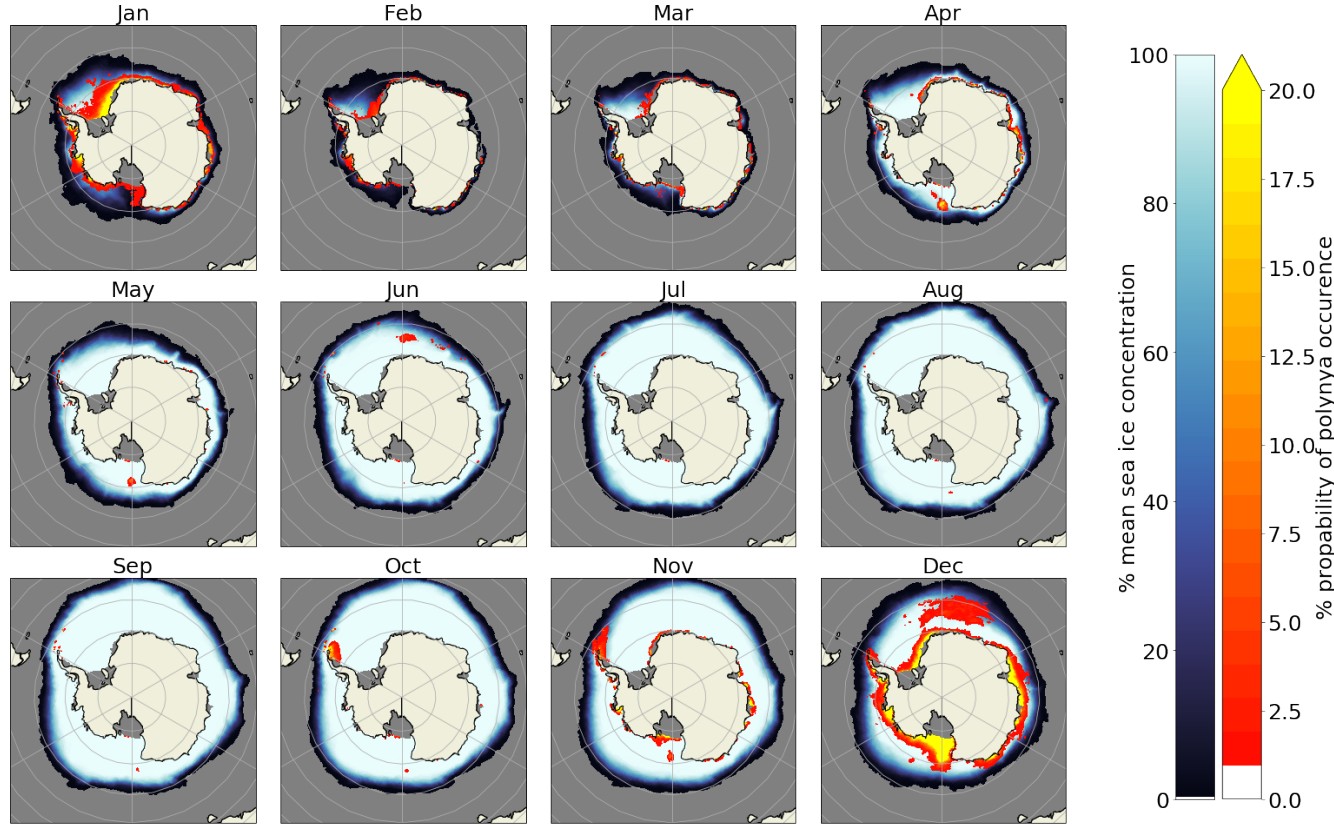

**Figure A5.** The seasonal cycle of sea ice concentration (blue colours) and polynya probability (red-yellow) for the GFDL-CM4 model.





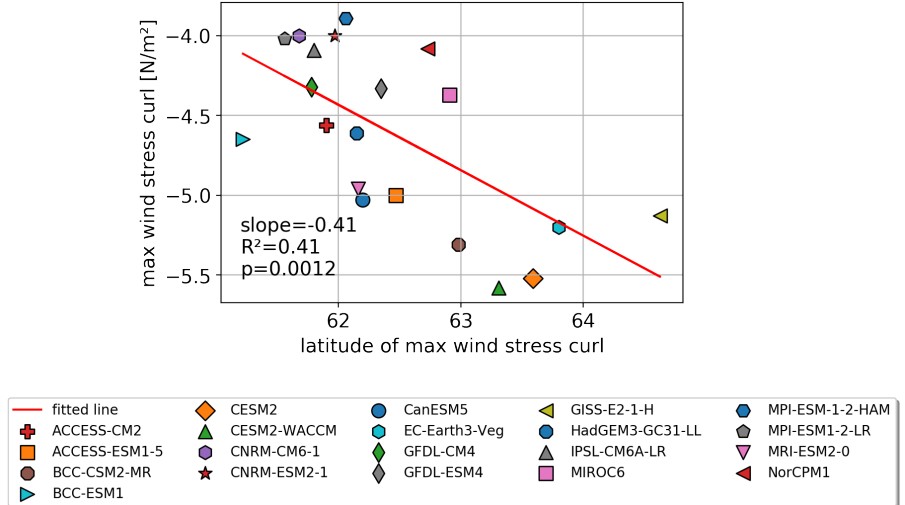

**Figure A6.** Linear regression plot of the maximum zonally integrated wind stress curl for each CMIP6 model and the latitude of that maximum. Values from Beadling et al. (2020).



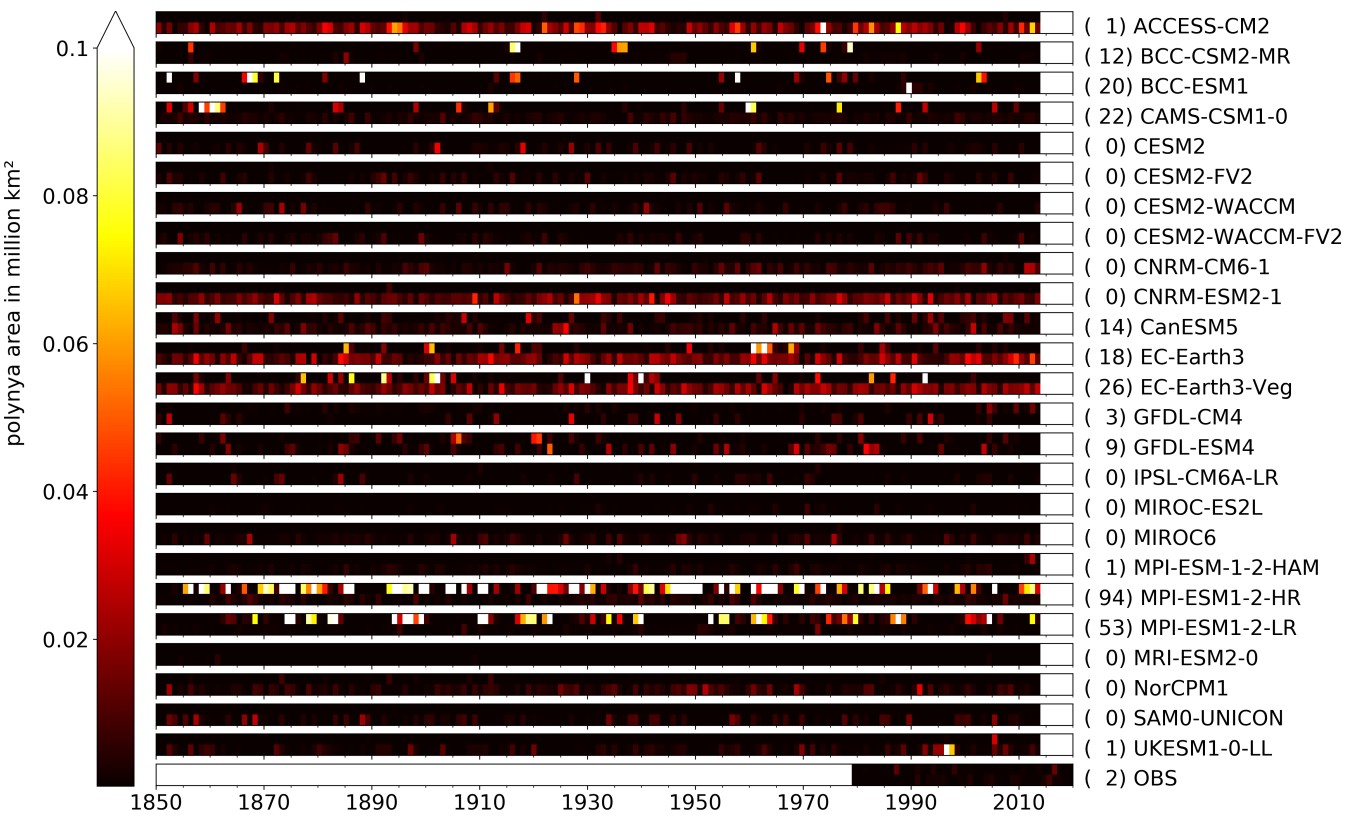

**Figure A7.** Yearly mean polynya area between May and November in historical CMIP6 model run for the Weddell Sea region, computed from monthly sea ice concentrations. Each bar represents one CMIP6 model and is divided horizontally into coastal polynya area (upper bar) and OWP area (lower bar). The values next to the model name represent the number of years an open water polynya with an area of more than $10.0 \times 10^3$ km$^2$ was detected.





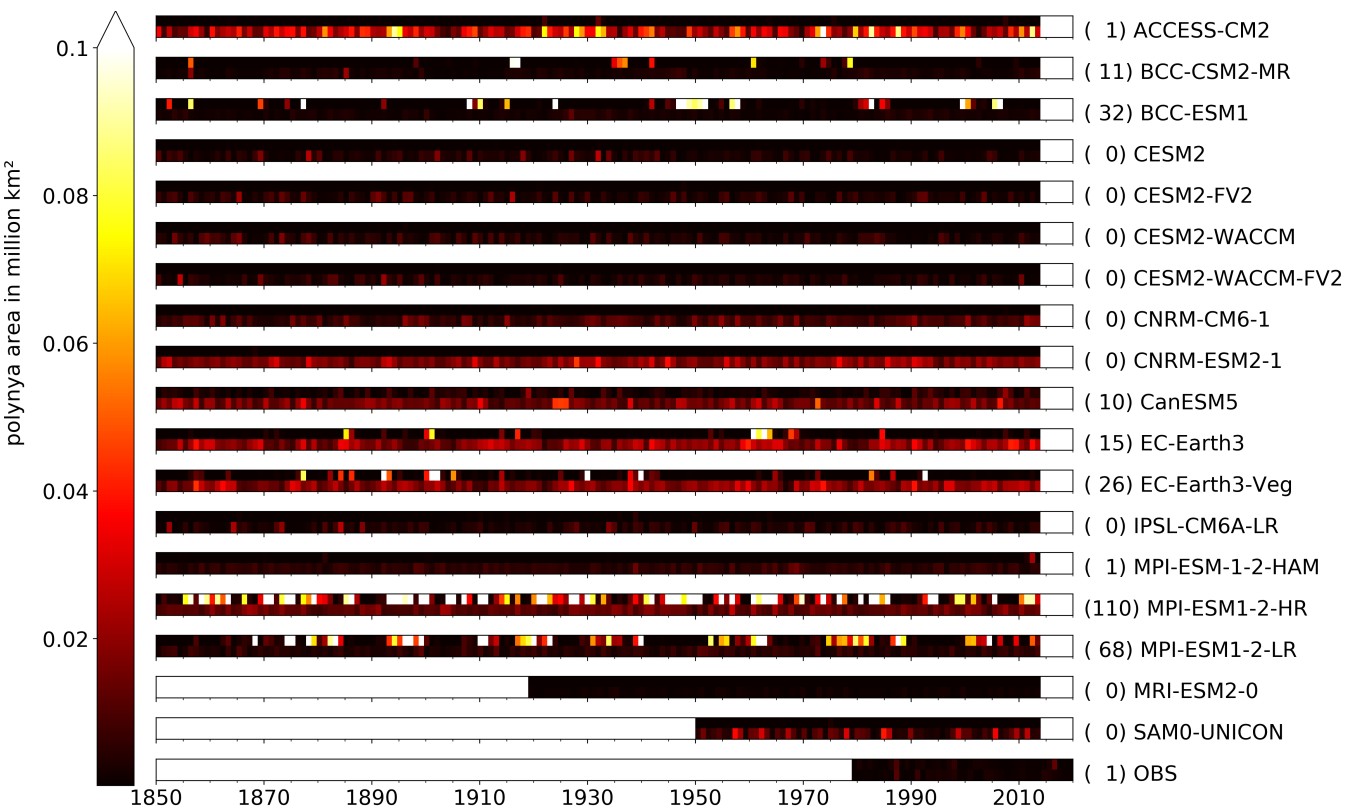

**Figure A8.** Yearly mean polynya area between May and November in historical CMIP6 model run for the Weddell Sea region, computed from daily sea ice concentrations. Each bar represents one CMIP6 model and is divided horizontally into coastal polynya area (upper bar) and OWP area (lower bar). The values next to the model name represent the number of years an open water polynya with an area of more than $10.0 \times 10^3$ km$^2$ was detected.



**Table A1.** The 28 CMIP6 models used in this study; full name and **references with model description**. The references marked with a "*" contain information about deep convection in the Southern Ocean. N/A indicates that no paper has been published yet for the CMIP6 configuration.

| Model Name | Descriptive Name | Reference |
| --- | --- | --- |
| ACCESS-CM2* | Australian Com. Clim. and Earth Syss Simulator Clim. Mod. Version 2 | Bi et al. (2013)* |
| ACCESS-ESM1-5* | Australian Com. Clim. and Earth Sys. Simulator Earth Sys. Model Ver 1.5 | Ziehn et al. (2017), Bi et al. (2013)* |
| BCC-CSM2-MR | Beijing Climate Center Climate System Model | Wu et al. (2019) |
| BCC-ESM1 | Beijing Climate Center Earth System Model | Wu et al. (2019) |
| CAMS-CSM1-0 | Chinese Academy of Met. Sciences Climate System Model 1.0 | Rong et al. (2019) |
| CESM2 | Com. Earth System Model ver. 2 | Danabasoglu et al. (2020) |
| CESM2-FV2 | Com. Earth System Model ver. 2 (Finite Volume) | Danabasoglu et al. (2020) |
| CESM2-WACCM | Com. Earth System Model ver. 2 (Whole Athm. Com. Clim. Model) | Danabasoglu et al. (2020) |
| CESM2-WACCM-FV2 | Com. Earth System Model ver. 2 (FV and WACCM) | Danabasoglu et al. (2020) |
| CNRM-CM6-1 | Centre National de Recherches Mét. - Clim. Model 6.1 | Voldoire et al. (2019) |
| CNRM-ESM2-1 | Centre National de Recherches Mét. - Earth Sys. Model 2.1 | Séférian et al. (2019) |
| CanESM5 | Canadian Earth System Model version 5 | Swart et al. (2019) |
| EC-Earth3 | European community Earth-System Model 3 | N/A |
| EC-Earth3-Veg | European community Earth-System Model 3 (Vegetation) | N/A |
| GFDL-CM4 | Geophysical Fluid Dynamics Laboratory - Climate Model 4 | Held et al. (2019) |
| GFDL-ESM4* | Geophysical Fluid Dynamics Laboratory - Earth System Model 4 | Dunne et al. (2019)* |
| GISS-E2-1-H* | Goddard Institute for Space Studies, ModelE atm. code, HYCOM | Kelley et al. (2019)* |
| HadGEM3-GC31-LL | Met Office Hadley Centre ESM | Kuhlbrodt et al. (2018) |
| IPSL-CM6A-LR* | Institut Pierre-Simon Laplace Clim. Modelling C. Clim. Model 6 Low Res. | Boucher et al. (2020)* |
| MIROC-ES2L | Model for Interdiscip. Res. on Clim. - Earth Sys. 2 for long term sim. | Hajima et al. (2020) |
| MIROC6* | Model for Interdiscip. Res. on Clim. | Tatebe et al. (2019)* |
| MPI-ESM-1-2-HAM* | Max-Planck-Institute-Earth System Model 1.2 + Aerosol sim | Mauritsen et al. (2019), Gutjahr et al. (2019)* |
| MPI-ESM1-2-HR | Max-Planck-Institute-Earth System Model 1.2-HR | Müller et al. (2018) |
| MPI-ESM1-2-LR* | Max-Planck-Institute-Earth System Model 1.2-LR | Mauritsen et al. (2019), Gutjahr et al. (2019)* |
| MRI-ESM2-0 | Meteorological Research Institute - Earth System Model 2.0 | Yukimoto et al. (2019) |
| NorCPM1 | Norwegian Climate Prediction Model version 1 | Counillon et al. (2016) |
| SAM0-UNICON | SNU Atm. Model ver. 0 with Unified Convection Scheme | Park et al. (2019) |
| UKESM1-0-LL* | U.K. Earth System Model 1.0 | Sellar et al. (2019)* |



*Author contributions.* MM, CH and SS designed the study. MM conducted the analyses under the supervision of CH and SS. MM and CH wrote the paper.

*Competing interests.* The authors declare no competing interest.

*Acknowledgements.* We are thankful for the comments of Dr. David Schroeder on a previous version of this manuscript. We acknowl-
edge the support from the Knut J:son Mark Foundation for financial support. CH is funded by the Swedish National Space Agency (dnr 164/18) and the Swedish Research Council (dnr 2018-03859).Sebastiaan Swart acknowledges support from the following grants: Wallen-berg Academy Fellowship (WAF 2015.0186), Swedish Research Council (VR 2019-04400), STINT-NRF Mobility Grant and NRF-SANAP (SNA170522231782).

We acknowledge the World Climate Research Programme, which made this paper possible by coordinating and promoting CMIP6. We
thank the climate modeling groups listed in Table 1 and supplementary table A1 for producing, unifying and providing their output, the Earth System Grid Federation (ESGF) for archiving the data and providing access, and the multiple funding agencies who support CMIP6 and ESGF. Data were collected and made freely available by the Southern Ocean Carbon and Climate Observations and Modeling (SOCCOM) Project funded by the National Science Foundation, Division of Polar Programs (NSF PLR-1425989), supplemented by NASA, and by the International Argo Program and the NOAA programs that contribute to it. The Argo Program is part of the Global Ocean Observing System
(https://doi.org/10.17882/42182, http://argo.jcommops.org).



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
