# Peer review of "Southern Ocean polynyas in CMIP6 models"

_The Cryosphere, 2021_

## Referee Comment (RC2)

**Southern Ocean polynyas in CMIP6 models**

submitted to the Cryosphere by Martin Mohrmann et al.

This paper examines the representation of coastal and open ocean polynyas (OWPs) in the Southern Ocean in the new generation of climate models that participate to the Climate Model Intercomparison Project phase 6 (CMIP6). The authors use satellite observations to assess the main characteristics (e.g. occurrence, area, location) of polynyas in 27 CMIP6 models owing to a detection algorithm based on sea ice concentration or thickness that the authors have developed and present for the first time in the paper. Temperature and salinity profiles from an Argo float is also used to assess the representation of stratification in the Weddell Sea region under "normal" conditions versus episodes of OWP. The authors find that only half of the 27 models form OWPs and that these polynyas are underestimated compared to observations. In contrast, all models simulate coastal polynyas but the associated area is overestimated. Links between winds and polynyas, as well as between the Antarctic Circumpolar Current and polynyas are explored. Finally, the authors provide a discussion on the strategies adopted by modelling centers to reduce the occurrence of spurious polynyas in models.

The topic of this paper fits well within the stated scope of the Cryosphere and provides a valuable evaluation of Southern Ocean polynyas in the new generation of climate models. This is a subject of importance given the impact of polynyas on regional and global climate, and given the challenges the representation of these events pose to the modelling community. The paper introduces a new algorithm used for the systematic detection of polynyas in models and observations that can prove to be useful to the polar community. The paper presents a thorough evaluation of polynya characteristics, discusses a substantial amount of material, and is generally well-presented with clear figures. I have quite many major and detailed comments which should all be easy to address (see below). I feel that addressing these comments would improve the quality of the manuscript and make it suitable for publication.

**Main comments**

*Introduction*
I feel that there are some elements about the representation of OWPs in models missing in the introduction that could be added before the question or objective of the study can be stated in the last paragraph. For instance, it is well known that the modelling community struggles with the opening of spurious large OWPs as is reported in the paper through the analysis of CMIP6 models and discussed at the end. The formation of OWPs in models has been shown to be very sensitive to vertical mixing schemes (e.g. Kjellson et al., 2015; Heuzé et al., 2015), initial sea ice conditions (e.g. Kjellson et al., 2015), stratification (e.g. De Lavergne et al., 2014, Stössel et al. 2015), and model resolution (Dufour et al. 2017, Kurtakoti et al. 2018, Lockwood et al. 2021), among other things. The value of the present study is to document the representation of OWPs in the new generation of climate models and assess whether major issues still remain. I suggest the authors add a paragraph on the modelling challenges related to the representation of OWPs. This will help bring the main question addressed in the study.

*Method*
- How does the algorithm deal with embayments? Many OWPs become embayments as they disappear. Some models also produce large embayments with associated deep convection, very much like fully closed polynyas. Accounting or not for embayments might have a significant effect on the results (number of polynyas, areas, life time of a fully closed polynyas when it evolves into an embayment, etc). Please comment on that.
- The detection algorithm based on sea ice thickness uses 'sivol' which is the sea ice volume per grid cell area. This is not the actual thickness but rather an equivalent thickness as you explain. Have you compared the equivalent thickness (sivol) with the actual thickness (sithick) for those models which do not show the weird behaviour shown in Figure 4? It seems to me that the two variables could give significantly different results.

*Results*
- Modelled ocean properties are compared to that taken from one Argo float from the SOCCOM project which conveniently sampled the 2016-2017 Maud Rise polynya. The profiles taken during the Maud Rise polynya are very relevant to the present study. But I am concerned about performing the comparison based on only one float. What about also using existing climatologies that cover a larger domain to evaluate the stratification "under sea ice" (e.g. See de Lavergne et al., 2014)?
- The manuscript does not say much about the trajectory and time period covered by the float. More information is needed (e.g. coordinates of domain covered, time period, number of profiles inside and outside the polynya). Furthermore, it is unclear to me whether the comparison between the models and the float is only performed over the Maud Rise. It seems that models are averaged over the Weddell Sea but I am not sure about the float. I suggest the authors clarify that in the text so that differences between models and observations are easier to understand.
- The difference between polynya and under sea ice profiles is much smaller in observations than in models (Fig. 10). In addition, we do not see a shallower subsurface temperature maximum during polynyas in the observations as we do in the models. Please comment on that.

*Discussion*
- Caution should be taken when inferring causes of differences between climate models (CMs) and Earth System models (ESMs). CMs and ESMs are different in many aspects (including additional model components but not only). Given that 1/ There are only a couple of CM/ESM pairs that can be compared (that belonging to the same family) across CMIP6 and 2/ Occurrence of polynyas is very sensitive to stratification (among other factors), it is hard to be make a conclusive and general statement on the representation of polynyas in CMs versus ESMs.
- Have you investigated the role of the representation of convection (convection schemes used in models) on the formation of OWPs? If relevant, information about convection schemes could be added in Table 1 for instance.

*Language*
- Please check the use of acronyms (e.g. OWP, SAM) as they are used inconsistently across the manuscript.

**Detailed comments**

***Abstract***
- p.1, l.7: I suggest you change "The coastal polynya area in contrast is often overestimated" to "In contrast, the coastal polynya area is overestimated in most models."
- p.1, l.10: I do not believe that introducing the ACC acronym is useful at this stage since it is not used in the abstract.
- p.1, l.13: What is meant by "vertical discretisation"? Do you mean new or more adequate vertical discretisation (for e.g. hybrid vertical coordinate system)?

***1. Introduction***
- p.1, l.23-34: Replace the "/" by "or".
- p.2, l.3-31: The statement about the influence of coastal polynyas on intermediate water formation only applies to the northern hemisphere I believe. Please verify this statement, and if it only applies to the northern hemisphere remove it as it is confusing.
- p.2 , l.34-35: It is unclear whether the statement applies to coastal polynyas, to open water polynyas, or to both. Please specify.
- p.2, l.35: "The Weddell Sea Polynya has been the largest OWP to date" → "The Weddell Sea Polynya has been the largest OWP **observed** to date".
- p.2, l.38: "More than forty years later, in 2016 and 2017, the Maud Rise polynya re-opened": This sounds like the Maud Rise polynya had never reopened before. I think what made the 2016-2017 exceptional is the size of the polynya as mentioned next in the sentence. Please rephrase to convey more clearly that point.
- p.2, l.40: I believe you mean "Weddell Sea and Maud Rise polynyas" by "OWPs" (I do not think the statement applies to all OWPs).
- p.2, l.40-41: How about the hypothesis made by Gordon et al. (2007) on the role of the transition of the Southern Annular Mode from a prolonged negative phase to a positive phase in explaining the Weddell Sea polynya of the 1970s? Even if you choose to keep the sentence as is I would add the reference to Gordon et al. (2007) to the other references.
- p.2, l.41-43: The correlation between OWPs and SAM/westerlies suggest a causality (winds driving the opening of OWPs). The correlation between open ocean convection and the ACC strength also suggests a causality, but the other way round: open ocean convection strengthens the meridional density gradient which in turn strengthens the ACC transport. I find the sentence about correlation between OWPs and ACC ambiguous in the sense that it does not say clearly what this correlation suggests. The reader could infer that the ACC transport might be driving the opening of the OWPs which I do not think is what Behrens et al. (2016) argued.
- p.2, l.45: "open water polynyas" should be "OWPs" for consistency.
- p.2, l.48: Are you really going to determine the causes of the OWPs for each of the 27 models? This seems to me unrealistic as causes are often multiple and complex requiring an in-depth analysis of a simulation which seems hard to carry out for all these models in one paper.
- p.2, l.58: "open water polynyas" should be "OWPs" for consistency.
- p.2, l.54-55: "We also analyse indicators of deep convection and upwelling as possible causes for the polynya formation and maintenance in CMIP6 models." I feel this is a

bit of an overstatement as Section 4.3 presents results on stratification but does not investigate causes. The diagnostics presented do not really allow that.

- p.2, l.55: Do "polynyas" refer to "OWPs" only or to both coastal and open water polynyas?

**2. CMIP6 output fields and observational data**

- p.4, l.61-76: I suggest you put that paragraph in a subsection instead of having it as a long introductory paragraph to Section 2.

- p.4, l.62-63: Please define acronyms and specify what type of data OSI-450 and SMOS are (observational/reanalysis? In situ/remote?)

- p.4, l.66: "was" → "is".

- p.4, l.70-71 & 75-76: "For the assessment of polynya activity in the Southern Ocean, we use sea ice concentration and sea ice thickness data. " This sentence appears twice.

- p.4, l.83-87: That answers my comment above about the nature of the observational products used. I still think you should briefly state what type of products OSI is in the previous section, as the reader who is unfamiliar with these products will keep wondering until reading this part of Section 2.1.

- p.5, l.96-97: It would be worth mentioning that sithick is the actual thickness.

- p.5, 100: "would be" → "is".

- p.5, l.101: "but we found it not good for our polynya detection": Could you be a bit more specific? "not good" is a bit vague. E.g. do you have false positive when using this variable?

- p.5, l.109: "that the form on the sea ice" → "that form on sea ice".

- p.5, l.112: Please remove "relatively new " as you already mention it a couple of sentences ago.

- p.5, l.116-117: This sentence belongs to the end of Section 2.1 as the advantages and disadvantages are discussed in that section and not in the present section.

- p.5, l.121: "rescaled" → " rescale to"?

**3. Polynya detection**

- p.6, l.141: "but not by its sea ice concentration" → "but not if using the threshold sea ice concentration criteria"?

- p.6, l. 141-142: Same comment as above. The phrasing sounds a bit off to me.

- p.6, l.150: " until the first polynya areas merge with the open ocean": Do you mean that the algorithm detects an embayment instead of a closed area of open ocean within the sea ice pack? On figure 2.b., we see that only coastal polynyas are detected (as we also see in fig 2e) for a concentration threshold of 60%. However, if the threshold further increases, then the OWP re-emerges. Could you please explain why?

- p.6, l.151-152: This seems to hold for the 60% threhsold in your example, but how about higher thresholds? I understand that the polynya would be detected but would be counted as smaller than when using a threshold smaller than 60%. Do you see similar behaviours in models (e.g. a particularly poor detection around the 60% threshold)? Please comment further on that.

- p.6, l.152: This seems to hold only until ∼85% based on Fig. 2b.

- p.6, l.153: Add "in" before "most".

- p.8, l.161-162: Please specify which "data" you are referring to. Observational data as shown in Fig. 2? Please also specify what "comparison" you are referring to (between sampling frequency)?

- p.8, l.163 (equation 1): Please describe what $A$ and $N$ refer to, as well as what each subscript refers to right after the equation. I am not sure to understand why you used the subscript "p" instead of "d". I thought this was referring to "daily".
- p.8, l.164: Do you mean "**coastal** polynyas grow very large in summer season"? There is little sea ice in Antarctica away from the coasts in summer to allow for a OWP to grow.
- p.9, l.172: "km$^3$" should be "km$^2$".
- p.9, l.173: Could it also be due to the fact that smaller polynyas would tend to live a shorter time than larger polynyas, hence have higher chances to be detected from daily output than from monthly output, while large polynyas persists for months, so would be equally as easy to detect using monthly or daily output?
- p.9, l.179-181: Would you have an explanation for that?
- Have you looked at the effect of sea ice model complexity (e.g. number of sea ice thicnkess categories) on the level of agreement between the sea ice concentration and sea ice thickness based algorithm?
- p.9, l.190 (equation 2): Should "$A_p$" be "$A_m$" in the second equation?
- p.10, l.204-206: Add the reference to Fig.4.
- p.10, l.205-209: What is the cause of this issue? I do not think it is expected to have ∼1 m of ice near the sea ice edge.
- p.10, l.226-227: I am not sure what you mean by "We have seen this". Do you refer to deep convection on the shelf? Please clarify.
- p.11, l.239-243: This paragraph needs a transition with the previous paragraph or a sentence of introduction as it is a bit disconnected from the rest.

**4. Polynya statistics in CMIP6**
- p.11, l.246: "evaluate" → "compare".
- p.11, l.251-252: I do not see a coastal polynya in the Ross Sea in e.g. BCC- or CAMS models. Is it because this polynya opens later than September so is not visible in either Figures?
- p.11, l.255-257: Agreed, but note the systematic overestimation in the model. Could it be due to a lack of resolution (since 0.25° is still too coarse for the coastal region)?
- p.11, l.259: You can also refer to Fig. 3.
- p.11, l.264-265: This is an interesting point. I presume ocean properties and circulation, and atmospheric forcing are fairy similar within each model family? Have you looked?
- p.12, l.268: "by" → "with".
- p.13, l.285: " 21 concentration" → "21 when using concentration".
- p.13, l.285-286: Isn't it redundant with l.272-273?
- p.13, l.286: Please add a comma after "Surprisingly".
- p.13, l.299: "they form" → "as they form".
- p.13, l.300: "We believe that ...". This statement needs to be supported by analyses. Please refer to a figure or to analyses performed.
- p.14, l.306: Have you looked at the frequency of occurrence in these models?
- p.15, l.319: There are two "the".
- p.15, l.325: "inside an open water polynya compared to the stratification under sea ice" → "inside and outside an OWP." Some OWP might host very thin ice.
- p.15, l.326: What is the northernmost latitude of the Weddell Sea domain?
- p.16, l.330-332: Do you average the float profiles over the Maud Rise only or do you

take into account all profiles regardless of the region? In any case, please indicate the coordinates of the domain covered by the float as well as the time.
- p.16, l.334: "is resulting" → "results".
- p.16, l.339: Please add a comma after "(Fig. 10)".
- p.16, l.346-347: Please clarify what you mean in the parentheses. The available sub-surface heat is mostly replenished by the CDW I believe.
- p.16, l.350: I am not sure what is referred to as "Active mixing processes". Rather "deep convection"?
- p.16, l.352-353: The float does not descend deeper than 2 km so please rephrase.
- p.16, l.353: "mixing" → "convection".
- p.16, l.357-358: Please clarify how this sentence links with the previous one.
- p.16, l.359: What does show similar mixing? The measurements from the 1970s Polynya? Grammatically, the subject of "shows" is "discussion".
- p.16, l.366: "is transporting" → "transports".

**5. Discussion**
- p.18, l.376: Please remove "will".
- p.18, l.376: " overestimation of OWPs in many models" seems at odds with "with the majority underestimating OWP area".
- p.18, 382-384: I am not sure I follow the argument. de Lavergne et al. (2014) also includes ESMs in the subset of models they analyze. Their conclusion about the cessation of the convection holds in general but there might be some significant differences across models, and between CMs and ESMs. Please clarify.
- p.19, l.402: 0.4 is a weak correlation. Please comment on that.
- p.19, l.309: Double comma in the parentheses.
- p.19, l.421-425: Alternatively, as most models simulate an ACC weaker than observations (presumably because of the misrepresentation of bottom water formation from coastal processes and spurious mixing leading to a too weak meridional density gradient), strong (overestimated) deep convection associated with OWPs compensate for that low bias through the formation of bottom waters that "correct" for the weak meridional density gradients. The sum of the two biases would lead to an improved representation of the strength of the ACC.
- p.19, l.414-425: In contrast to the winds, it appears that the strength of the ACC is not a cause for the occurrence of polynyas, but rather a consequence of large polynyas. I think this point should be made more clear.
- p.20, l.432: "general known" → "general knowledge"?
- p.20, l.443-444: See Lockwood et al. (2021) for a discussion on the role of coastal freshening on the occurrence of OWPs.
- p.20, l.459-460: I feel that Figure 11 could be moved to the Appendix as it is not a major figure of the paper.
- p.21, l.470-472: You might find Adcroft et al. (2019) useful for this part of the discussion (e.g. see their Section 3.8.3). The introduction of the new hybrid coordinate system in MOM6 did not solve as much as hoped.

**6. Conclusion**
- p.21, l.477: "best agreement" with observations?
- p.21, l.485: Please remove the reference to the section.
- p.21, l.485: Please specify what you refer to by "upwelling" (upward mixing of water resulting from gravitational instability or something else)? I do not recall this was specifically mentioned/discussed in the results section (the word "upwelling" does not appear there).
- p.21, l.487-488: There is only one paragraph in Section 5.3 on the comparison between the two ACCESS models, so I do not think that sentence summarizes the main point of that section.

**Data availability**
- Please provide the link for SOCCOM.

**Figures**

*Figure 1*
- Please specify in the caption what grey color corresponds to.

*Figure 2*
- I think you should add a dark blue color in your colorbar corresponding to open water.
- The colorbar is discrete but we see shades of blue around the sea ice edge or polynya edges. I assume this is an effect of the filter. Please add a note in the caption.

*Figure 4*
- Please explain what the two colorbars each correspond to (The caption only says "polynya areas").
- Please explain what the grey color corresponds to. Open ocean north of the sea ice edge and ice shelves both appear in grey.
- Perhaps adding a contour of the sea ice edge on panel j) would help visualize the issue described by the authors.

*Figure 6*
- Please add a reference to the appropriate section and equations in the third sentence of the caption.
- The fact that some models do not have daily output or equivalent thickness leads to unequal length of the x-axis across panels a) to c) which makes comparing models across the different panels difficult. To remedy that, you could keep the full list of models on the x-axis for all panels with no data for models which miss output. I believe that would make things clearer.

*Figure 8*
- "open water polynya" → "OWP".

*Figure 9*
- Which domain did you took for the Weddell Sea? Please provide the range of longitudes and latitudes.
- The labels of months on the x-axis are really packed. Maybe you could display one label every other label or rotate the labels?
- I am not convinced that the use of symbols for displaying models is better than the use of plain and dashed lines with different colors. Symbols make the shape of the plot hard to grasp.

*Figure 10*
- Please indicate in the caption the change of vertical scale between the first two and last two columns.
- There are some ".8" and "-0.4" here and there on the figure that should be removed I believe.
- Does the T-S diagram use in situ or potential temperature? Which flavour of the density is used? Please specify.
- Please specify the number of profiles for the float inside and outside the polynya (in the caption or in the text). It does not seem that there are many profiles inside the polynya.
- I suggest cutting the profiles at 2 km for the models to ease the comparison with the observations. The details of the Argo float profiles are hard to see and the most interesting features appear over the top 1 km for both models and observations anyway.

*Table 2*
- I think keeping three significant digits for all variables (especially for sa_tot that could be written in $10^6$ km$^2$) is enough. Also, that will make the numbers easier to read and compare.

*Figure A1*
- The figure is of poor resolution. Could you please try to improve that?
- I suggest you color-code the numbers in the legend (e.g. red for the coastal polynyas). That will be easier to read.

*Figure A2*
- Same comments as in Fig. A1.

**References**

Adcroft, A., Anderson, W., Balaji, V., Blanton, C., Bushuk, M., Dufour, C. O., et al. (2019). The GFDL global ocean and sea ice model OM4.0: Model description and simulation features. Journal of Advances in Modeling Earth Systems, 11, 3167-3211. https://doi.org/10.1029/2019MS001726´.

Heuzé, C., Ridley, J. K., Calvert, D., Stevens, D. P., and Heywood, K. J.: Increasing vertical mixing to reduce Southern Ocean deep convection in NEMO3.4, Geosci. Model Dev., 8, 3119-3130, https://doi.org/10.5194/gmd-8-3119-2015, 2015.

Kjellsson, J., and Coauthors, 2015: Model sensitivity of the Weddell and Ross Seas, Antarctica, to vertical mixing and freshwater forcing. Ocean Modell., 94, 141-â152, doi:10.1016/j.ocemod.2015.08.003.

Kurtakoti, P., Veneziani, M., Stössel, A., & Weijer, W. (2018). Preconditioning and Formation of Maud Rise Polynyas in a High-Resolution Earth System Model, Journal of Climate, 31(23), 9659-9678.
https://journals.ametsoc.org/view/journals/clim/31/23/jcli-d-18-0392.1.xml.

Lockwood, J. W., Dufour, C. O., Griffies, S. M., & Winton, M. (2021). On the Role of the Antarctic Slope Front on the Occurrence of the Weddell Sea Polynya under Climate Change, Journal of Climate, 34(7), 2529-2548.
https://journals.ametsoc.org/view/journals/clim/34/7/JCLI-D-20-0069.1.xml.

Stössel, A., D. Notz, F. A. Haumann, H. Haak, J. Jungclaus, and U. Mikolajewicz, 2015: Controlling high-latitude Southern Ocean convection in climate models. Ocean Modell., 86, 58-75, doi:10.1016/j.ocemod.2014.11.008. https://doi.org/10.1016/j.ocemod.2014.11.008.

---

## Author Response (AR1)

**Reply to reviewers comments on "Southern Ocean polynyas in CMIP6 models", submitted to The Cryosphere by Martin Mohrmann, Céline Heuzé and Sebastiaan Swart**

We thank both reviewers for their careful reading of our paper and the proposed improvements. We are convinced that the additional clarifications will increase the readability and scientific value of our paper. Here, we are addressing the comments of anonymous reviewer 1. We acknowledged the importance of the reviews in the acknowledgment section:

"We thank the anonymous reviewer, Carolina Dufour and Rebecca Beadling for their comments, which greatly helped us improve the quality of our writing and frame the presented research in a wider scientific context."

**1. Comment from Anonymous Referee #1 and our responses**

This manuscript examines the representation of polynyas in CMIP6 models as compared to observations. Some of these comparisons are not straightforward, due to lack of CMIP model variables, limited observations, and the different metrics that could be used to define polynyas, but the authors are transparent in these limitations and convey the information clearly. Modelled coastal polynyas are often too large, likely as a result of coarse horizontal resolution. Modelled open water polynyas are often too small compared to observations, and there is a large inter-model spread in the frequency of open water polynyas. The authors examine vertical ocean profiles in polynyas versus sea ice covered regions in a subset of the models and in float data. The Discussion contains a number of useful insights on the reasons behind the intermodel variation in polynya activity, relating to resolution, simulation of the ACC and overflow parametrizations.

I found this to be a very interesting and thorough paper. It is well within the scope of TC and presents novel results and conclusions. The methods are clearly explained and the analysis code has been made publicly available. It is generally well-written, apart from some of the latter sections, and the figures and tables are appropriate. I am selecting 'major revisions' only because of section 5.2, which I think would benefit from a second round of reviews.

**1.1 Main comments**

Section 5.2 - I found the arguments here a bit hard to follow. I would like to see additional subplots for the other relationships discussed here added to Figure A6. As you mention the results from this section in the abstract and conclusions, the figure should also be brought into the main paper. It seems like the results here would be interesting to a wide audience, so I think it is worth spending some more time on the presentation.

Thank you for this suggestion. We added the discussed relationships to the Figure A6 and brought it into the main paper. We also added some additional explanations to Section 5.2.

Section 4.3 or Section 2.3 - Please give some more details on the domain of the SOCCOM float - e.g. time period, number of profiles etc. Please also describe how you extracted the profiles from the CMIP6 models - is this one profile per grid cell in the Weddell Sea region? Is there some time averaging?

We agree that this information was lacking and have now added a more detailed description of our method:

"We concentrate on the models that form most OWPs (see Table 2) and show only the top three models (MPI-ESM1.2-HR, ACCESS-ESM1.5, BCC-ESM1) in Fig. 10. For comparison, we present the observed hydrographic data of a SOCCOM profiling float (Johnson et al., 2018), which was deployed in January 2015 and surfaced two times in the Maud Rise Polynya in winter 2017 (Campbell et al., 2019). To provide regionally and seasonally comparable data sets for the models and the profiling float, we chose to extract vertical profiles during the month of September from within a rectangle around the profiling float trajectory (see Campbell et al., 2019) with the edge coordinates 61°S-66°S, 0°E-6°E in the Weddell Sea. This region includes the northern flank of Maud Rise, where we found OWPs to be most common (Fig. 3, A3, A4).

For the SOCCOM profiling float, we use the information provided in Campbell et al. (2019) to differentiate vertical profiles when the float surfaces within an open water polynya from those sampled under the sea ice. For the models, we use our algorithm to differentiate and group the grid points by whether they are within an OWP or not. Based on this criteria, we extract and group the vertical salinity and conservative temperature profiles (monthly) and plot them in either blue (under sea ice) or red color (OWP) in Figure 10."

L329: 'To evaluate the effect of OWPs on vertical stratification' - 'To evaluate vertical stratification in OWPs' (also L371) - as there isn't a clear cause and effect relationship here.

Changed as suggested

Conclusions - I would like to see more of the polynya statistics summarised here. This could work well as a bulleted list.

Our conclusions now start with a bullet list summarizing important polynya statistics and findings:

"In this paper, we evaluated the representation of Southern Ocean open water and coastal polynyas in CMIP6 climate models and their effects on the modelled Weddell Sea. We found that:

      All 27 analysed models have coastal polynyas around the Antarctic continent, while OWPs are present in only half of the models

      CMIP6 models show OWPs most commonly in either the Weddell or the Ross Seas

      The position of polynya formation is very similar for models of the same family and likely determined by the model properties

      In comparison to observations, nine models underestimate polynya areas based on thickness threshold but overestimate them if based on concentration threshold method

      Coastal polynyas in CMIP6 have a large annual variability of at least a factor of 2.5

      With total polynya areas from 6.5 x 10^3 km2 up to 215 x 10^3 km2, CMIP6 models show a large intermodel spread"

**1.2 Minor comments**

L8 'presence or absence of OWPs are' > 'presence or absence of OWPs is'

→ changed as suggested

L12 'that require to be addressed' > 'that should/must be addressed'

→ changed to "should be addressed"

L30 requires citation

→ added required citation (Tamura et al, 2008)

L85 Suggest adding a sentence on uncertainty in SIC observations

→ Added as suggested.

"The uncertainty in sea ice concentration is less than 4% on average (Lavergne et al., 2019)."

Table 1 - please add units on R_o and R_a (otherwise a very nice table though!)

added ([km]) (and thank you for the compliment)

L101 'not good' > 'poor'

→ changed as suggested

L134 Please describe what a 'flood fill algorithm' is

We added a brief description and a citation of the used python-library.

"With the aim of detecting polynyas, we start with the sea ice concentration or thickness (Fig. 1a). To mask out the open ocean beyond the northern sea ice extent, we use a "flood-fill" algorithm from the scikit-image library (Van der Walt et al., 2014). Starting from a grid cell with no sea ice, the seed, the algorithm detects similar cells below a specified sea ice concentration/thickness and masks them out, effectively "filling them" with ice (Fig. 1b). Afterwards, a maximum sea ice threshold filter returns all grid cells that are classified as polynyas".

Fig. 3 'propability' > 'probability' on colorbar. I would also make all of the ocean dark blue (not grey).

We corrected the spelling in this Figure and found and corrected the same error in related Figures.

We discussed making the ocean dark blue in all the figures of the paper. While it would improve the clarity of the Figures, there is unfortunately no (easy) consistent way to mark the ice shelves in the CMIP6 models in another color than the ocean, as these areas are not differentiated in the sea ice output. In practice that means that we cannot easily color the ocean dark blue without coloring the ice shelves and (or) continent dark blue for some of the models. We estimate making the suggested color change work for all models would take some days of work on a relatively small visual only benefit, so we would prefer to leave it as it is.

Fig. 4 - shouldn't this be 'equivalent ice thickness' not 'floe thickness'?

In Figure 4 we show that the floe thickness (CMIP6 variable name: sithick) cannot be used with our algorithm and therefore, we continue our analysis with the sea ice concentration and equivalent ice thickness variables (siconc and sivol). We find this naming scheme somewhat confusing, but want to stay as close as possible to the terminology used in the CMIP6 guidelines and documentation.

Fig. 6 'All data sets where' > 'All data sets were'; 'its' full length' > 'its full length'

Corrected

Fig. 8 (and similar figures in the appendix). I like this visualisation, but I wonder if you can separate the coastal and open water polynya bars and make the whole figure taller so it is easier to see?

We separated the color bars for OWPs and coastal polynyas with some white space in between and made all three figures taller.

L163 - Why doesn't the mean of daily data go from 1st May to the end of Nov?

During the ice melting phase in late November, observed and modeled polynyas often become an order of magnitude larger than during the winter (Figure 9). We did not want these polynyas to dominate our results. Moreover, when we provide a comparison of the maximum yearly polynya area (computed according to Eq. 2) in Figure 6 (transparent bars), these maximum values often reflect mainly the large November polynyas, and no further time averaging within the season is done. We consider a comparison between the averaged values of November (Figure 6a) and the daily values of the 15th November (Figure 6b) more accurate than including values from the end of November and comparing those with values derived from the monthly averaged data.

L334 'is resulting in' > 'results in'

→ changed as suggested

L340 'Compared to the float data, ACCESS and BCC underestimate…'

→ changed as suggested

L353 'deeper reaching' -> 'deeper-reaching' ?

→ changed as suggested

L354: 'There are some profiles' - please be more quantitative

changed to: About half of the profiles have a shallower and, for the MPI model, warmer temperature maximum compared to the under ice case.

L366: Define N^2 in the text

changed to "the Brunt-Väisälä frequency squared (N²)..."

Fig. 10 - Please add subplot labels (a, b, c, …) and refer to these in the main text. This will make the text easier to follow

We added subplot labels and the corresponding references in the text.

L383 'This is consistent…' Rephrase, it's not clear here what you mean

changed to:

"The decreased OWP activity we find for CMs in our CMIP6 dataset with ongoing global warming is consistent with the results of de Lavergne et al. (2014), in which OWPs in the Weddell Sea eventually stop at the end of the extended CMIP5 climate change runs, as the CM models show a stronger warming response than the ESM-versions (Dong et al., 2020)."

L386 'can usually be run'; remove 'in our case'

changed as suggested

L438 'All these parameters are positively correlated with OWP activity **in observations**'

added as suggested

**Citation**: https://doi.org/10.5194/tc-2021-23-RC1

**2. Additional changes**
Changes that are not listed in this document can be found in our response to Carolina Dufour. Moreover, we received a helpful suggestion aside from the public review process, which we also want to address here. A reader of our preprint pointed out that we counted the total number of CMIP6 models in Table 2 of the cited Beadling et al. 2020 paper incorrectly, and that it would be preferable to stress which results were significant. Thank you for the corrections, we improved this part:

L436-438
Before: "However, Beadling et al. (2020) found that 30 of 35 CMIP6 models underestimated the ACC, 34 of 38 showed their wind stress curl minimum not sufficiently south and 33 of 38 underestimated the wind stress curl maximum. All these parameters are positively correlated with OWP activity (Campbell et al., 2019)"

Improved: "However, Beadling et al. (2020) found that *out of the 34 CMIP6 models they analysed, 29 underestimated the ACC (of which only 12 significantly), 30 show their wind stress curl not sufficiently south (5 significantly) and 30* underestimated the WSC minimum (9 significantly)."

**Referenced in our responses (for further references see manuscript):**

- Beadling, R., Russell, J., Stouffer, R., Mazloff, M., Talley, L., Goodman, P., Sallée, J., Hewitt, H., Hyder, P., and Pandde, A.: Representation of Southern Ocean Properties across Coupled Model Intercomparison Project Generations: CMIP3 to CMIP6, Journal of Climate, 33, 6555–6581, 2020.
- Griffies, S. (2015). A handbook for the GFDL CM2-0 model suite.
- Hasumi, Hiroyasu. "CCSR ocean component model (COCO)." (Version 4.0) (2015).
- Heuzé, C.: Antarctic Bottom Water and North Atlantic Deep Water in CMIP6 models, Ocean Science, 17, 59–90, 2021.
- Jungclaus, J. H., et al. "Characteristics of the ocean simulations in the Max Planck Institute Ocean Model (MPIOM) the ocean component of the MPI-Earth system model." *Journal of Advances in Modeling Earth Systems* 5.2 (2013): 422-446.
- Keen, Ann, et al. "An inter-comparison of the mass budget of the Arctic sea ice in CMIP6 models." *The Cryosphere* 15.2 (2021): 951-982.
- Madec, G., Bourdallé-Badie, R., Bouttier, P. A., Bricaud, C., Bruciaferri, D., Calvert, D., ... & Vancoppenolle, M. (2017). NEMO ocean engine.
- Van der Walt, S., Schönberger, J. L., Nunez-Iglesias, J., Boulogne, F., Warner, J. D., Yager, N., Gouillart, E., and Yu, T.: scikit-image: image processing in Python, PeerJ, 2, e453, 2014.

**Reply to reviewers comments on "Southern Ocean polynyas in CMIP6 models", submitted to The Cryosphere by Martin Mohrmann, Céline Heuzé and Sebastiaan Swart**

We thank both reviewers for their careful reading of our paper and the proposed improvements. We are convinced that the additional clarifications will increase the readability and scientific value of our paper. Here we are addressing the comments of Carolina Dufour. We acknowledged the importance of the reviews in the acknowledgment section:

"We thank the anonymous reviewer and Carolina Dufour for their comments, which greatly helped us improve the quality of our writing and frame the presented research in a wider scientific context."

**1. Comment from Carolina Dufour and our responses**

This paper examines the representation of coastal and open ocean polynyas (OWPs) in the Southern Ocean in the new generation of climate models that participate to the Climate Model Intercomparison Project phase 6 (CMIP6). The authors use satellite observations to assess the main characteristics (e.g. occurrence, area, location) of polynyas in 27 CMIP6 models owing to a detection algorithm based on sea ice concentration or thickness that the authors have developed and present for the first time in the paper. Temperature and salinity profiles from an Argo float is also used to assess the representation of stratification in the Weddell Sea region under "normal" conditions versus episodes of OWP. The authors find that only half of the 27 models form OWPs and that these polynyas are underestimated compared to observations. In contrast, all models simulate coastal polynyas but the associated area is overestimated. Links between winds and polynyas, as well as between the Antarctic Circumpolar Current and polynyas are explored. Finally, the authors provide a discussion on the strategies adopted by modelling centers to reduce the occurrence of spurious polynyas in models. The topic of this paper fits well within the stated scope of the Cryosphere and provides a valuable evaluation of Southern Ocean polynyas in the new generation of climate models. This is a subject of importance given the impact of polynyas on regional and global climate, and given the challenges the representation of these events pose to the modelling community. The paper introduces a new algorithm used for the systematic detection of polynyas in models and observations that can prove to be useful to the polar community. The paper presents a thorough evaluation of polynya characteristics, discusses a substantial amount of material, and is generally well-presented with clear figures. I have quite many major and detailed comments which should all be easy to address (see below). I feel that addressing these comments would improve the quality of the manuscript and make it suitable for publication.

We thank the reviewer for their detailed and constructive comments, which we have all addressed as detailed in the following pages.

**1.1 Main comments**

**Introduction**

I feel that there are some elements about the representation of OWPs in models missing

in the introduction that could be added before the question or objective of the study can be stated in the last paragraph. For instance, it is well known that the modelling community struggles with the opening of spurious large OWPs as is reported in the paper through the analysis of CMIP6 models and discussed at the end. The formation of OWPs in models has been shown to be very sensitive to vertical mixing schemes (e.g. Kjellson et al., 2015; Heuzé et al., 2015), initial sea ice conditions (e.g. Kjellson et al., 2015), stratification (e.g. De Lavergne et al., 2014, Stössel et al. 2015), and model resolution (Dufour et al. 2017, Kurtakoti et al. 2018, Lockwood et al. 2021), among other things. The value of the present study is to document the representation of OWPs in the new generation of climate models and assess whether major issues still remain. I suggest the authors add a paragraph on the modelling challenges related to the representation of OWPs. This will help bring the main question addressed in the study.

We think your framing of the problems in modelling OWPs is adding great value to the introduction, so we included it without too many changes:
*"The formation of OWPs in models has been shown to be very sensitive to vertical mixing parameters (e.g. Kjellsson et al., 2015; Heuzé, 2015), initial sea ice conditions (e.g. Kjellsson et al., 2015), stratification (e.g. De Lavergneet al., 2014; Stoessel et al., 2015), and model resolution (e.g. Dufour et al., 2017; Kurtakoti et al., 2018; Lockwood et al.,2021). Spurious OWP appearance or deep convection in the Weddell Sea remain a challenge in modern climate models (e.g. Held et al., 2019; Sellar et al., 2019; Mauritsen et al., 2019, and references marked with a '\*' in Table A1). A weak background stability, especially in the Weddell Sea (Wilson et al., 2019), makes the ocean susceptible to convective overturning due to model inaccuracies, such as a lack of dense shelf water overflows (Dufour et al., 2017) or heat buildup due to insufficient vertical mixing (Heuzé et al. 2015). Ocean convection due to static instabilities is an important process in the formation of OWPs, which is not modeled directly due to the relatively coarse resolution of many CMIP6 models but parameterised instead (e.g. Hasumi, 2000; Madec et al., 2017). Despite the crucial role of OWPs for sea ice production and deep water properties, an evaluation and comparison of their representation across current climate models is missing."*

**Method**
- How does the algorithm deal with embayments? Many OWPs become embayments as they disappear. Some models also produce large embayments with associated deep convection, very much like fully closed polynyas. Accounting or not for embayments might have a significant effect on the results (number of polynyas, areas, life time of a fully closed polynyas when it evolves into an embayment, etc). Please comment on that.
We modified section 3.1 to discuss the handling of embayments:
*"The influence of different sea ice thresholds on our four step algorithm is visualised in Fig. 2: the areas that are classified as polynyas increase with higher sea ice concentration thresholds, until the first polynya areas merge with the open ocean and thus become embayments (Fig.2b,e). High sea ice concentration thresholds (up to ~85%) maximize the number of detectable coastal polynyas, but lead to poor recognition of OWPs (Fig. 2e). The higher the chosen sea ice threshold, the lower the area that is classified as sea ice."*

Moreover, we added a new sentence to the Caveats:

*"Large embayments from the open ocean into the sea ice can reach latitudes and areas comparable to those of OWPs, especially in the melting season. We use a strict definition of polynyas as areas surrounded by sea ice and do not account or compensate the results for eventual embayment areas."*

- The detection algorithm based on sea ice thickness uses 'sivol' which is the sea ice volume per grid cell area. This is not the actual thickness but rather an equivalent thickness as you explain. Have you compared the equivalent thickness (sivol) with the actual thickness (sithick) for those models which do not show the weird behaviour shown in Figure 4? It seems to me that the two variables could give significantly different results.

As we mentioned in section 3.3, we find this unusual behavior for the majority of the models.
There are eight models left that provide daily sea ice floe thickness, which we did not analyse previously (EC-Earths, CNRMs, MIROCs).
Of these, the MIROCs are lacking equivalent sea ice thickness so the suggested comparison cannot be done. We will concentrate our comparison on the EC-Earths and CNRMs models.
The EC-Earths provide a somewhat similar output for the equivalent sea ice thickness and the floe thickness. Note however, that equivalent thickness is only provided in a monthly averaged form, while the floe thickness is provided daily, so we computed the monthly time average ourselves for the comparison that follows.

[Figure]

[Figure]

From its definition, the equivalent thickness, which is volume divided by area, must be lower (or equal) at all times than the ice floe thickness.. So in theory, we would detect larger polynyas with the equivalent sea ice thickness data than with the sea ice floe thickness data. As we have already seen in the paper with other models, we find anomalies of very thick sea ice floe thickness at the outer sea ice boundaries for the model EC-Earth3, see e.g. the region north of the Antarctic Peninsula in the upper left corner. We already show These anomalies led to poor results in our polynya detection algorithm. To highlight this, we found your comment below concerning our papers' Figure 4 helpful and included a 1% sea ice concentration isoline in that figure.

For the CNRMs, we find reasonable values for the daily sea ice floe thickness. Since these are the only two models that provide daily sea ice floe thickness and do not have too heavy biases, we do not add their comparison to the study, so that our message remains clear: the floe thickness output is heavily biased and needs to be fixed.

**Results**
- Modelled ocean properties are compared to that taken from one Argo float from the SOCCOM project which conveniently sampled the 2016-2017 Maud Rise polynya. The profiles taken during the Maud Rise polynya are very relevant to the present study. But I am concerned about performing the comparison based on only one float. What about also using existing climatologies that cover a larger domain to evaluate the stratification "under sea ice" (e.g. See de Lavergne et al., 2014)?

We now included data from the EN4 subsurface temperature and salinity dataset.
"*In comparison to the CMIP6 models, the data from the SOCCOM float is of short length (2015-2017) and limited depth (maximum 2000 m). We hence use additional under sea ice profiles from the EN4.2.1 full-depth ocean temperature and salinity data set (Good et al., 2013), limiting it to the same spatial domain as for the CMIP6 models and show the spatially averaged profiles from 1980-2015 in light grey color in Fig. 10m-p. We provide the*

*EN4.2.1 data as additional reference of the observed water stratification without major OWP events*"

- The manuscript does not say much about the trajectory and time period covered by the float. More information is needed (e.g. coordinates of domain covered, time period, number of profiles inside and outside the polynya). Furthermore, it is unclear to me whether the comparison between the models and the float is only performed over the Maud Rise. It seems that models are averaged over the Weddell Sea but I am not sure about the float. I suggest the authors clarify that in the text so that differences between models and observations are easier to understand.

We agree that this information was lacking and added a more detailed description of our method. Note that we also recomputed Figure 10 to be sure that it complies with our updated description:

"*We concentrate on the models that form most OWPs (see Table 2) and show only the top three models (MPI-ESM1.2-HR, ACCESS-ESM1.5, BCC-ESM1) in Fig. 10. For comparison, we present the observed hydrographic data of a SOCCOM profiling float (Johnson et al., 2018), which was deployed in January 2015 and surfaced two times in the Maud Rise Polynya in winter 2017 (Campbell et al., 2019). To provide regionally and seasonally comparable data sets for the models and the profiling float, we chose to extract vertical profiles during the month of September from within a rectangle around the profiling float trajectory (see Campbell et al., 2019) with the edge coordinates 61°S-66°S, 0°E-6°E in the Weddell Sea. This region includes the northern flank of Maud Rise, where we found OWPs to be most common (Fig. 3, A3, A4).*
*For the SOCCOM profiling float, we use the information provided in Campbell et al. (2019) to differentiate vertical profiles when the float surfaces within an open water polynya from those sampled under the sea ice. For the models, we use our algorithm to differentiate and group the grid points by whether they are within an OWP or not. Based on this criteria, we extract and group the vertical salinity and conservative temperature profiles (monthly), average spatially over the different vertical profiles to obtain one averaged under sea ice profile and one averaged OWP profile (if OWP present in domain) per time step.  We plot these profiles then in either blue (under sea ice) or red color (OWP) in Figure 10.*"

- The difference between polynya and under sea ice profiles is much smaller in observations than in models (Fig. 10). In addition, we do not see a shallower subsurface temperature maximum during polynyas in the observations as we do in the models. Please comment on that.

We have now added the following comment about this to the manuscript:
"*In comparison to Argo float data, the vertical mixing in the models seems overestimated, which was to be expected  as we here selected the subsample of three models with the highest OWP activity. Moreover, the float data contains only two profiles from the very edge of the 2017 Maud Rise Polynya event (Campbell et al., 2019), where the vertical mixing may be less prominent than at the polynya's centre. A discussion of the stratification changes from the 1970s Weddell Sea Polynya can be found in Cheon and Gordon (2019), where similar mixing as in the CMIP6 models was found.*"

**Discussion**
- Caution should be taken when inferring causes of differences between climate models (CMs) and Earth System models (ESMs). CMs and ESMs are different in many aspects (including additional model components but not only). Given that 1.) There are only a couple of CM/ESM pairs that can be compared (that belonging to the same family) across CMIP6 and 2.) Occurrence of polynyas is very sensitive to stratification (among other factors), it is hard to be make a conclusive and general statement on the representation of polynyas in CMs versus ESMs.

We agree that conclusive statements are not possible with so few models. We advise against generalisation now at the end of Section 5.1:

"*Our comparison between CM and ESM was only possible on the three out of fifteen model families that provided both types of model and as such may not apply to the other families.*"

- Have you investigated the role of the representation of convection (convection schemes used in models) on the formation of OWPs? If relevant, information about convection schemes could be added in Table 1 for instance.

Ocean convection due to vertical instabilities cannot be simulated by the CMIP6 models directly. One reason is the use of the hydrostatic approximation, which separates the direct link between buoyancy and vertical acceleration of water in the ocean Hasumi et al. (2015). The second reason is the relatively coarse horizontal resolution of the models (at best 25 km), which would not allow for the representation of convection cells even without the hydrostatic approximation. Two parameterisations of the convection are most common for CMIP6 models (e.g. Jungclaus et al. (2013), Hasumi et al. (2015), Madec et al. (2017)):

1. Non-penetrative convective adjustment. (HadGem3, MIROC6, MIROC-ESM1, MRI-ESM2.0, NorCPM1)
2. Enhanced vertical diffusion. (ACCESSs, BCCs, CESM2s, CNRMs, EC-Earths, GFDLs, IPSL-CM6A, MPIs, UKESM1.0LL)

Sources: https://view.es-doc.org/ or the respective model documentations.

The implementation of this convective schemes can vary, e.g. convective adjustment can be done pairwise or with more elaborate numerical schemes, a static instability can be removed by repeated application of pairwise convective adjustment in one timestep or it could be applied only once leading to a longer adjustment duration. In the second method, the parameter of vertical diffusivity is increased if the stratification is unstable.

Even though none of the models using non-penetrative adjustment has OWPs, Heuzé et al (2021) found that HadGEM3 and the MIROCs have major areas of open ocean convection in the Weddell- and Ross Seas. This is in agreement with de Lavergne et al. (2014), who found that the "pre-industrial presence or absence of deep convection does not relate to the numerical choices to parameterize open ocean convective processes". In summary we conclude that the choice of convection scheme does not significantly influence the formation of OWPs.

We add a line about the de Lavergne et al. (2014) findings to our manuscript:
"*In contrast to the stratification, numerical choices of parameterisation of open ocean convective processes were not significant for the presence of absence of deep convection in CMIP5 models (*de Lavergne et al., 2014)*.*"

**Language**
- Please check the use of acronyms (e.g. OWP, SAM) as they are used inconsistently across the manuscript.
→ checked and changed

**1.2 Detailed comments**

**Abstract**
- p.1, l.7: I suggest you change "The coastal polynya area in contrast is often overestimated" to "In contrast, the coastal polynya area is overestimated in most models."
→ changed as suggested

- p.1, l.10: I do not believe that introducing the ACC acronym is useful at this stage since it is not used in the abstract.
→ changed as suggested

- p.1, l.13: What is meant by "vertical discretisation"? Do you mean new or more adequate vertical discretisation (for e.g. hybrid vertical coordinate system)?
→ We mean for example isopycnal or terrain following coordinates as we show in Table 1 and discuss later in the paper. In the abstract, we now say more adequate vertical grid type.

**Introduction**
- p.1, l.23-34: Replace the "/" by "or".
→ changed as suggested
- p.2, l.3-31: The statement about the influence of coastal polynyas on intermediate water formation only applies to the northern hemisphere I believe. Please verify this statement, and if it only applies to the northern hemisphere remove it as it is confusing.
Thanks for the correction, we removed the statement.
- p.2 , l.34-35: It is unclear whether the statement applies to coastal polynyas, to open water polynyas, or to both. Please specify.
We meant OWPs; we now specify:
*"Some coastal polynyas open at the same location every year (e.g. the Ross Sea Polynya), while OWPs are observed only once per decade or less."*
- p.2, l.35: "The Weddell Sea Polynya has been the largest OWP to date" → "The Weddell Sea Polynya has been the largest OWP observed to date".
→ changed as suggested

- p.2, l.38: "More than forty years later, in 2016 and 2017, the Maud Rise polynya re-opened": This sounds like the Maud Rise polynya had never reopened before. I think what made the 2016-2017 exceptional is the size of the polynya as mentioned next in the sentence. Please rephrase to convey more clearly that point.
We rephrased:
*"Only minor polynyas have been observed in the region from 1976 to 2016 (Cheon et al, 2019), when the Maud Rise polynya (Swart et al, 2018; Heuzé et al, 2019; Francis et al, 2019) reached an area larger than 50 000 km², similar to the polynya of October-November 1973 (Cheon et al, 2019)."*

- p.2, l.40: I believe you mean "Weddell Sea and Maud Rise polynyas" by "OWPs" (I do not think the statement applies to all OWPs).

→ changed as suggested

- p.2, l.40-41: How about the hypothesis made by Gordon et al. (2007) on the role of the transition of the Southern Annular Mode from a prolonged negative phase to a positive phase in explaining the Weddell Sea polynya of the 1970s? Even if you choose to keep the sentence as is I would add the reference to Gordon et al. (2007) to the other references.
→ reference added as suggested.

- p.2, l.41-43: The correlation between OWPs and SAM/westerlies suggest a causality (winds driving the opening of OWPs). The correlation between open ocean convection and the ACC strength also suggests a causality, but the other way round: open ocean convection strengthens the meridional density gradient which in turn strengthens the ACC transport. I find the sentence about correlation between OWPs and ACC ambiguous in the sense that it does not say clearly what this correlation suggests. The reader could infer that the ACC transport might be driving the opening of the OWPs which I do not think is what Behrens et al. (2016) argued.
We reformulated to make the cause and consequence relationship clearer:
*"These Weddell Sea and Maud Rise polynyas were preceded by a phase of positive Southern Annular Mode (SAM) anomalies and strong Southern Hemisphere westerlies (Campbell et al., 2019; Cheon et al., 2014; Gordon et al 2007). Behrens et al. (2016) found that deep convection, as it happens in OWPs, is strengthening the meridional density gradients and increases the Antarctic Circumpolar Current (ACC) transport on multidecadal time scales."*

- p.2, l.45: "open water polynyas" should be "OWPs" for consistency.
→ changed as suggested

- p.2, l.48: Are you really going to determine the causes of the OWPs for each of the 27 models? This seems to me unrealistic as causes are often multiple and complex requiring an in-depth analysis of a simulation which seems hard to carry out for all these models in one paper.
We agree that statement was too strong.
We changed it: "Here we determine the characteristics,  and impacts of Southern Ocean polynyas in 27 models."

- p.2, l.58: "open water polynyas" should be "OWPs" for consistency.
→ changed as suggested

- p.2, l.54-55: "We also analyse indicators of deep convection and upwelling as possible causes for the polynya formation and maintenance in CMIP6 models." I feel this is a bit of an overstatement as Section 4.3 presents results on stratification but does not investigate causes. The diagnostics presented do not really allow that.
We removed this sentence, since we already discussed the possible impact of OWPs on the water stratification in the previous sentence.

- p.2, l.55: Do "polynyas" refer to "OWPs" only or to both coastal and open water polynyas?
We removed this sentence, see previous comment. We had only OWPs in mind though.

2. CMIP6 output fields and observational data
- p.4, l.61-76: I suggest you put that paragraph in a subsection instead of having it as a long introductory paragraph to Section 2.
As suggested, we moved these sentences to a new subsection entitled: "Choice of CMIP6 models"

- p.4, l.62-63: Please define acronyms and specify what type of data OSI-450 and SMOS are (observational/reanalysis? In situ/remote?)
→ changed as suggested

*"Routine satellite based sea ice concentration observations are available from January1979 onwards, so there are 35 years of overlap with the historical CMIP6 model run on which we can perform our comparisons.We use the daily, satellite observation based sea ice concentration product "Ocean and Sea Ice 450" (OSI-450) (Lavergne et al., 2019) for the time period 1979-2015." [...]*

*"For observational comparison of the sea ice thickness, we use the daily, satellite-based "Soil Moisture and Ocean Salinity" thin sea ice thickness product (Huntemann et al., 2014)."*

- p.4, l.66: "was" → "is".
→ changed as suggested

- p.4, l.70-71 & 75-76: "For the assessment of polynya activity in the Southern Ocean, we use sea ice concentration and sea ice thickness data. " This sentence appears twice.
→ thank you for the careful reading, we removed one of the sentences.

- p.4, l.83-87: That answers my comment above about the nature of the observational products used. I still think you should briefly state what type of products OSI is in the previous section, as the reader who is unfamiliar with these products will keep wondering until reading this part of Section 2.1.
We no longer provide the specific acronym in the first sentences of the methods section , but added a cross reference to the corresponding subsection instead.
*"We used for this study two remote observation based sea ice products for sea ice concentration and thickness that we introduce inSections 2.2 and 2.3, and the sea ice output from the historical run of 27 CMIP6 models (Eyring et al., 2016, listed in Table 1)."*

- p.5, l.96-97: It would be worth mentioning that sithick is the actual thickness.
Based on the experience we made with the sithick variable, reaching values of over 20 m in almost ice free areas etc and inside of polynyas (see Figure 4 and its discussion), we prefer to not call the sithick variable 'actual thickness' or 'actual ice floe thickness' at this point, since we believe that it does not reflect our or the readers intuitive expectation of 'ice floe thickness' yet.

- p.5, 100: "would be" → "is".
→ changed as suggested

- p.5, l.101: "but we found it not good for our polynya detection": Could you be a bit more specific? "not good" is a bit vague. E.g. do you have false positive when using this variable?
We changed "not good" to "poor". Moreover, we would like to give a specific reasoning already in this paragraph, but since we explain several biases in Section 3.3, we just refer

to that section to avoid using an oversimplified statement here. (e.g. MPI model have different bias then CESM models).
We also added some explanation to Section 3.3:
*"When using our thickness threshold algorithm with this dataset, we obtain large OWPs in areas that hardly correspond to the observations (Fig. 4a-i), because the high ice floe thickness anomalies at the outer sea ice edges encircle large areas with thinner ice floe thickness."*

- p.5, l.109: "that the form on the sea ice" → "that form on sea ice".
→ changed as suggested

- p.5, l.112: Please remove "relatively new " as you already mention it a couple of sentences ago.
→ changed as suggested

- p.5, l.116-117: This sentence belongs to the end of Section 2.1 as the advantages and disadvantages are discussed in that section and not in the present section.
→ changed as suggested (note that section 2.1 became 2.2 as we added a new subsection title above)

- p.5, l.121: "rescaled" → " rescale to"?
→ changed as suggested

3. Polynya detection
- p.6, l.141: "but not by its sea ice concentration" → "but not if using the threshold sea ice concentration criteria"?
→ changed as suggested

- p.6, l. 141-142: Same comment as above. The phrasing sounds a bit off to me.
→ changed as suggested

- p.6, l.150: " until the first polynya areas merge with the open ocean": Do you mean that the algorithm detects an embayment instead of a closed area of open ocean within the sea ice pack?
Yes, exactly. We changed this to:
*"...until the first polynya areas merge with the open ocean and thus become embayments. Embayments are not classified as polynyas."*

On figure 2.b., we see that only coastal polynyas are detected (as we also see in fig 2e) for a concentration threshold of 60%. However, if the threshold further increases, then the OWP re-emerges. Could you please explain why?
see our response to the next comment
- p.6, l.151-152: This seems to hold for the 60% threshold in your example, but how about higher thresholds? I understand that the polynya would be detected but would be counted as smaller than when using a threshold smaller than 60%. Do you see similar behaviours in models (e.g. a particularly poor detection around the 60% threshold)?

Please comment further on that.

If we increase the sea ice concentration threshold, it means that the areas detected as "sea ice" are decreasing. Thus the open ocean and polynyas are in fact growing in area. In this process, polynyas might become embayments and are thus counted as part of the open ocean. For our example, the big polynya in the Weddell Sea becomes an embayment (Fig 2e) at about 55% sea ice concentration threshold. Every sea ice concentration threshold higher than 55% will not bring back that polynya, it will continuously be an embayment.

The reason that OWP areas are increasing after the 60% threshold again in Fig. 2b, is that new areas e.g. in the Ross Sea or the Amundsen Sea will be detected as OWPs. See the tiny emerging OWPs in the Amundsen Sea in Fig. 2e. If we increase the threshold even further to ~85%, even the Amundsen Sea OWPs and most of the coastal polynyas will merge with the ocean and become embayments. However, the polynyas we detect for high thresholds are not always what we would expect intuitively (e.g. Amundsen OWPs) so we use the 30% threshold instead. We also modified the explanations in the paper to be clearer about the thresholds and embayments:

*"For thresholds higher than 60%, (Fig. 2b) the OWP area is increasing again after the big OWP in the Weddell Sea merged with the open ocean. However, this signal is relatively noisy in time (not shown) and sensitive to small threshold changes (Fig. 2b). While more and more polynyas become embayments, new areas with more than 60% sea ice concentration are classified as polynyas and the area classified as "sea ice covered" in between shrinks."*

- p.6, l.152: This seems to hold only until ~85% based on Fig. 2b.
Yes, that is right. We added this information in the text.
*"High sea ice concentration thresholds (up to ~85%) maximize the number of detectable coastal polynyas, but lead to poor recognition of OWPs."*

- p.6, l.153: Add "in" before "most".
→ changed as suggested

- p.8, l.161-162: Please specify which "data" you are referring to. Observational data as shown in Fig. 2? Please also specify what "comparison" you are referring to (between sampling frequency)?
Before: We transformed the data to a common time resolution to allow for a comparison.
Improved to: *"When comparing the daily observational data to monthly model output, we computed the monthly time mean of the sea ice concentration and thickness first and applied our algorithm then. To provide yearly averaged winter polynya area for each model and observational product, we used yearly polynya area averages: [...]"*

4- p.8, l.163 (equation 1): Please describe what A and N refer to, as well as what each subscript refers to right after the equation. I am not sure to understand why you used the subscript "p" instead of "d". I thought this was referring to "daily".
→ Thank you for your suggestions, the subscripts were indeed confusing before. We changed them to 'd' for daily and 'm' for monthly now. Moreover, we explain the introduced symbols below the formula.

*"In Equation 1, $A_d(t)$ and $A_m(t)$ are the polynya areas derived from one day/month. $N_{days}$ and $N_{months}$ are the total number of days/month in the time series."*

- p.8, l.164: Do you mean "coastal polynyas grow very large in summer season"? There is little sea ice in Antarctica away from the coasts in summer to allow for a OWP to grow.

Some models do not have a strong enough summer sea ice minimum (see Figure 9 or Figure 4k). However, it is mostly coastal polynyas that remain in the observations and (most) models after December, so we changed it to 'coastal polynyas'.

- p.9, l.172: "km³ " should be "km² ".
→ changed as suggested

- p.9, l.173: Could it also be due to the fact that smaller polynyas would tend to live a shorter time than larger polynyas, hence have higher chances to be detected from daily output than from monthly output, while large polynyas persists for months, so would be equally as easy to detect using monthly or daily output?
That's a good point, we included it in the text.
"*We suspect that smaller polynyas have a shorter average life span and are more prone to relative changes in area due to ice drift.*"

- p.9, l.179-181: Would you have an explanation for that?
Yes! We added an explanation to the text.
"*The observational sea ice thickness retrieval algorithm assumes sea ice concentrations of 100% (Hunteman, 2014), while the sea ice thickness variable we use for our analysis is an area averaged sea ice thickness and thus directly connected to changes in sea ice concentration.*"

- Have you looked at the effect of sea ice model complexity (e.g. number of sea ice thickness categories) on the level of agreement between the sea ice concentration and sea ice thickness based algorithm?
The BCC, CESM, GFDL, IPSL and MRI models have the lowest agreement (low R-values and slopes) in the linear correlations (Figure 2a). We compiled a table with sea ice model parameters based on the official CMIP6 documentations (https://explore.es-doc.org/cmip6) and (Keen et al, 2019) which includes all models which submitted sea ice information to es-doc.

| Model Name | Sea ice model component | Sea ice model resolution | Rheology | No of thickn. categ. | Radiation scheme | Thermodynamics |
|---|---|---|---|---|---|---|
| BCC-CSM2/ESM1 | SIS (MOM) | Same as ocean | EVP | 5 | | Sentner |
| HadGEM3-GC31-LL | CICE 5.1.2 (GSI8.1) | same as ocean | EVP | 5 | Dual Band (CCSM3) | BL99 |
| HadGEM3-GC31-MM | CICE 5.1.2 (GSI8.1) | same as ocean | EVP | 5 | Dual Band (CCSM3) | BL99 |
| UKESM1-0-LL | CICE 5.1.2 (GSI8.1) | same as ocean | EVP | 5 | Dual Band (CCSM3) | BL99 |

| | | | | | | |
|---|---|---|---|---|---|---|
| EC-Earth-3 | LIM3 | same as ocean | EVP | 5 | Broadband | BL99 |
| EC-Earth3-Veg | LIM3 | same as ocean | EVP | 5 | Broadband | BL99 |
| MRI-ESM2 MRI 5 | MRI.COM 4.4 | same as ocean | EVP | 5 | Dual Band (CCSM3) | MK89 |
| CESM2-CAM | CICE 5.1.2 | same as ocean | EVP | 5 | Delta-Eddington | T13 |
| CESM2-WACCM | CICE 5.1.2 | same as ocean | EVP | 5 | Delta-Eddington | T13 |
| GFDL-CM4 | SIS2 | same as ocean | EVP | 5 | Delta-Eddington | BL99 |
| GFDL-ESM4 GFDL 1 | SIS2 | same as ocean | EVP | 5 | Delta-Eddington | BL99 |
| ACCESS-CM2 | CICE 5.1.2 (GSI8.1) | same as ocean | EVP | 5 | Dual Band (CCSM3) | BL99 |
| NorESM2-LM | CICE 5.1.2 | same as ocean | EVP | 5 | Delta-Eddington | T13 |
| NorESM2-MM | CICE 5.1.2 | same as ocean | EVP | 5 | Delta-Eddington | T13 |
| CanESM5 | LIM2 | same as ocean | EVP | 1 | Mulit-Band | 0-layer |
| IPSL-CM6A-LR | LIM3 | same as ocean | EVP | | | Energy con. Halo-therdyn. |

The above-mentioned models use the model sea ice components SIS, CICE, SIS2, LIM3, MRI.COM4.4 respectively, all have five sea ice thickness categories.
Based on the table, we cannot see a clear connection between the used sea ice component or its number of sea ice categories and our polynya areas based on thickness/ concentration thresholds.

- p.9, l.190 (equation 2): Should "A_p " be "A_m " in the second equation?
→ Yes, changed to $A_m$ as suggested

- p.10, l.204-206: Add the reference to Fig.4.
→ added as suggested

- p.10, l.205-209: What is the cause of this issue? I do not think it is expected to have ~1 m of ice near the sea ice edge.
We do not know where that issue comes from. Possibly the newly introduced variable "sea ice floe thickness" was just computed using pre-existing variables for sea ice concentration and equivalent thickness by the modeling groups. The different biases that we describe in

Section 3.3 suggest multiple issues with different causes. Investigating them in more detail would require access to the models' code and is beyond the scope of this paper.

- p.10, l.226-227: I am not sure what you mean by "We have seen this". Do you refer to deep convection on the shelf? Please clarify.

We replaced "this" with a specific explanation:

"For example, according to our analysis, IPSL-CM6A-LR has few OWPs but mostly coastal polynyas; yet these *coastal polynyas reach far out of the continental shelf and this model was reported to undergo deep convection (Heuzé et al, 2020), which suggests a misclassification of its OWPs as coastal polynyas*."

- p.11, l.239-243: This paragraph needs a transition with the previous paragraph or a sentence of introduction as it is a bit disconnected from the rest.

We now merged two paragraphs where we talk about spatial and temporal biases to increase readability:

"*A possible negative bias in our polynya area estimations could be sea ice embayments. In some models, areas of open water that are still connected to the open ocean can stretch far into the sea ice and reach latitudes and areas comparable to those of OWPs, especially in the melting season. We use a strict definition of polynyas as areas surrounded by sea ice and do not account or compensate the results for eventual embayment areas. Another bias could be the choice of one ensemble member for each model. In several models (GFDL-, ACCESS-, UKESM1-0-LL) multi-centennial variability causes periods with high polynya activity followed by times with few (Sellar et al, 2019). The ACCESS-ESM1-5 shows a long term warming trend of more than 1°C over the 165 model years (not shown). The BCC models have a strong multidecadal variance in sea ice extent (Beadling et al, 2020) which also affects the polynya areas (Section 4.3). In these models with long term variability, the results for the polynya activity could be less representative because we derive them from one ensemble member only which might coincidentally be in a phase of high or low polynya activity.*"

4. Polynya statistics in CMIP6
- p.11, l.246: "evaluate" → "compare".

→ changed as suggested

- p.11, l.251-252: I do not see a coastal polynya in the Ross Sea in e.g. BCC- or CAMS models. Is it because this polynya opens later than September so is not visible in either Figures?

Yes, exactly. We mention these models explicitly in the sentence now:

"*We here evaluate the modelled location and spatial extent of coastal and open water polynyas from the sea ice concentration and thickness variables. All 27 CMIP6 models exhibit coastal polynyas. The largest coastal polynya, the Ross Sea polynya, is represented in all models (Fig. 3, A3, A4), although some models open the Ross Sea polynya later than September (BCCs, CAMS-CSM1).*"

- p.11, l.255-257: Agreed, but note the systematic overestimation in the model. Could it be due to a lack of resolution (since 0.25 ∘ is still too coarse for the coastal region)?

We believe that a high resolution (higher than 0.25°) can improve the accuracy of polynya locations, that is the general trend we observed as mentioned in Section 5.1. A low resolution does not seem to lead to an overestimation of polynyas in general.

We note the overestimation of the model in the text now:

*"Even though GFDL-CM4 overestimates the polynya areas, the shapes and occurrence probability, for example of the Ross Sea polynya or the polynyas along the East Antarctic coast, are similar."*

- p.11, l.259: You can also refer to Fig. 3.
→ Good idea, added

- p.11, l.264-265: This is an interesting point. I presume ocean properties and circulation, and atmospheric forcing are fairly similar within each model family? Have you looked?
The Beadling et al. (2020); Keen et al. (2020); and Heuzé et al. (2021) papers show that models within one family behave relatively similar in their Southern Ocean properties.

- p.12, l.268: "by" → "with".
→ changed as suggested

- p.13, l.285: " 21 concentration" → "21 when using concentration".
→ changed as suggested

- p.13, l.285-286: Isn't it redundant with l.272-273?
Yes, we removed the  l.285-286
- p.13, l.286: Please add a comma after "Surprisingly".
→added as suggested
- p.13, l.299: "they form" → "as they form".
→ changed as suggested

- p.13, l.300: "We believe that ...". This statement needs to be supported by analyses. Please refer to a figure or to analyses performed.
We backed this up with some references:
*"We believe that the deep convection is the reason for the OWPs, because the models that have OWPs mention convective overturning in the Southern Ocean in their model documentation(Table A1) and the water stratification within the polynyas indicates convection (Section 4.3)."*
- p.14, l.306: Have you looked at the frequency of occurrence in these models?
In line 306, we talk about  MPI, ACCESS, BCC, CAMS-CSM1 and EC-Earth models. Two lines below we already describe the frequency of polynya events in some of these models:
*"While ACCESS-CM2 and BCC-(ESM1/CSM2) show OWPs almost every year, MPI-ESM1.2-(LR/HR), EC-Earth3 and CAMS-CSM1 show episodes of high OWP activity separated by multiple decades of little to no OWPs."*
Further information is available in Figure 8, A7 and A8 that we already refer to.
- p.15, l.319: There are two "the".
→ fixed
- p.15, l.325: "inside an open water polynya compared to the stratification under sea ice" → "inside and outside an OWP." Some OWP might host very thin ice.
→ changed as suggested

- p.15, l.326: What is the northernmost latitude of the Weddell Sea domain?
We specified the domain more in detail now:
*"To provide regionally and seasonally comparable data sets for the models and the profiling float, we chose to extract vertical profiles during the month of September from within a rectangle around the profiling float trajectory (Campbell et al, 2019) with the edge*

*coordinates 61°S-66°S, 0°E-6°E in the Weddell Sea. This region includes the northern flank of Maud Rise, where we found OWPs to be most common.*

- p.16, l.330-332: Do you average the float profiles over the Maud Rise only or do you take into account all profiles regardless of the region? In any case, please indicate the coordinates of the domain covered by the float as well as the time.
We added a new paragraph providing this information:
*"For the SOCCOM profiling float, we use the information provided in Campbell et al. (2019) to differentiate vertical profiles when the float surfaces within an open water polynya from those sampled under the sea ice. For the models, we use our algorithm to differentiate and group the grid points by whether they are within an OWP or not.*
*Based on this criterion, we extract and group the monthly vertical salinity and conservative temperature profiles and average spatially over the different vertical profiles to obtain one averaged under sea ice profile and one averaged under OWP profile (if OWP present in domain) per time step. We plot these profiles then in either blue (under sea ice) or red color (OWP) in Figure 10. In comparison to the CMIP6 models, the SOCCOM float data is of limited length (2015-2017) and depth (max 2000 m). To back the data up with more under sea ice profiles, we use the EN4.2.1 analyses subsurface ocean temperature and salinity (Good et al., 2013), limited to the same spatial domain as for the CMIP6 models and plot the profiles from 1980-2015 in light grey color in Fig. 10 m-p."*

- p.16, l.334: "is resulting" → "results".
→ changed as suggested

- p.16, l.339: Please add a comma after "(Fig. 10)".
→ changed as suggested

- p.16, l.346-347: Please clarify what you mean in the parentheses. The available sub-surface heat is mostly replenished by the CDW I believe.
We find it interesting that this MPI model (Figure 10b) has some profiles that are homogenous in depth and very cold while other profiles have a distinct, relatively shallow heat maximum . The parenthesis aimed to suggest a mechanism that could explain this distinct maximum. We rephrased:

*Old version: "The coldest profiles within polynyas are in fact the least stratified ones. For the model MPI-ESM1.2-HR (Fig 10a-d) for example, the under sea ice temperature varies about 0.5°C from the mean value, but varies by more than 1°C inside OWPs. A possible explanation is that there is available heat upon the opening of polynyas (e.g. due to Ekman pumping, warm profiles)."*

*Improved version: "The coldest profiles within polynyas are in fact the least stratified ones. For the model MPI-ESM1.2-HR (Fig 10a-d) for example, the under sea ice temperature varies about 0.5°C from the mean value, but varies by more than 1°C inside OWPs. A possible explanation could be that e.g. wind driven upwelling contributed to a shallowing of the MLD (hence the comparatively shallow heat maximum), leading to increased entrainment of warm CDW into the mixed layer and the opening of an OWP. Upon the opening of the polynya, the water cools rapidly while deep convection occurs, which could explain why the coldest profiles are the least stratified ones. The monthly resolution of the datasets is too low for us to verify these hypotheses."*

- p.16, l.350: I am not sure what is referred to as "Active mixing processes". Rather

"deep convection"?

→ Yes, we changed this to deep convection which is the most likely mixing process to happen here.

- p.16, l.352-353: The float does not descend deeper than 2 km so please rephrase.

"*In comparison to Argo float measurements, which we found to have stable stratification below 500 m even in the polynya profiles, the models show deeper-reaching convection with a portion of the profiles showing no substantial stratification to the bottom (5000 m).*"

- p.16, l.353: "mixing" → "convection".

changed as suggested

- p.16, l.357-358: Please clarify how this sentence links with the previous one.

Before: "*In comparison to Argo float data, the vertical mixing in the models seems overestimated, which was expected because we chose the models with the highest OWP activity. Moreover, the float data features only a few profiles from the outer edge of one Maud Rise Polynya event (Campbell et al., 2019).*"

Improved: "*In comparison to Argo float data, the vertical mixing in the models seems overestimated, which was to be expected as we here selected the subsample of three models with the highest OWP activity. Moreover, the float data features only two profiles from the very edge of the 2017 Maud Rise Polynya event (Campbell et al., 2019), where the vertical mixing might be less prominent than in the middle of the polynya.*"

- p.16, l.359: What does show similar mixing? The measurements from the 1970s Polynya? Grammatically, the subject of "shows" is "discussion".

→ we rephrased

- p.16, l.366: "is transporting" → "transports".

→ changed as suggested

**Discussion**

- p.18, l.376: Please remove "will".

→ changed as suggested

- p.18, l.376: " overestimation of OWPs in many models" seems at odds with "with the majority underestimating OWP area".

changed

- p.18, 382-384: I am not sure I follow the argument. de Lavergne et al. (2014) also includes ESMs in the subset of models they analyze. Their conclusion about the cessation of the convection holds in general but there might be some significant differences across models, and between CMs and ESMs. Please clarify.

Our original chain of arguments was this: De Lavergne argues that deep convection will stop eventually due to a warming climate. Dong et al. (2016) shows that climate warming is stronger in climate models than in ESM models. According to these two studies then, because ESMs have less warming, they keep deep convection longer and show more OWPs, which can explain what we observe.

We realize that this cannot be generalized for two reasons:

1. We analyse only three examples of models that allow for a comparison (as you mentioned previously)
2. In the study of Lavergne et al. (2014), we find two CMIP5 models that stop deep convection earlier than their CM counterparts.

We added this information to our discussion:
*"Our comparison between CM and ESM was only possible on the three out of fifteen model families that provided both types of model and as such may not apply to the other families. In contrast to our results, de Lavergne et al. (2014) found that deep convection in some CMIP5 models ceases earlier than in ESM models, but likewise their analysis was limited to only three families of models.*

- p.19, l.402: 0.4 is a weak correlation. Please comment on that.
While we were replotting the figure in response to the comments of Reviewer 1, we noticed that the original plot only contained 20 models instead of the 23 indicated in Section 5.2. By adding the data points that were missing, the Pearson correlation coefficient increased to 0.64.
Your comment is still valid, this is still not a very strong correlation. However, it is significant with a p value smaller than 0.01. It seems that a positive SAM index does not increase the wind stress curl equally much in all models, there must be more influence factors.

- p.19, l.309: Double comma in the parentheses.
→Thanks, we removed the double comma.

- p.19, l.421-425: Alternatively, as most models simulate an ACC weaker than observations (presumably because of the misrepresentation of bottom water formation from coastal processes and spurious mixing leading to a too weak meridional density gradient), strong (overestimated) deep convection associated with OWPs compensate for that low bias through the formation of bottom waters that "correct" for the weak meridional density gradients. The sum of the two biases would lead to an improved representation of the strength of the ACC.
We added the following:
*"Since most models simulate an ACC weaker than the observations (Beadling et al., 2020), an explanation could be that the models suffer in general from insufficient deep water formation, which is weakening the meridional density gradients and vertical stratification, but in turn lead to OWPs in some models that partially compensate for the lack of dense bottom water with spurious deep convection (Heuzé, 2015)."*

- p.19, l.414-425: In contrast to the winds, it appears that the strength of the ACC is not a cause for the occurrence of polynyas, but rather a consequence of large polynyas. I think this point should be made more clear.
See previous comment.

- p.20, l.432: "general known" → "general knowledge"?
→ changed as suggested

- p.20, l.443-444: See Lockwood et al. (2021) for a discussion on the role of coastal freshening on the occurrence of OWPs.
→Thanks, we included that paper in our discussion.

*"One of the development foci for the GFDL models was to reduce Southern Ocean polynyas (Held et al., 2019). For the GFDL-CM2 models, the formation of super polynyas in the Southern Ocean was addressed by increasing the near-infrared albedo of glaciers*

*and snow covered ice caps in order to increase coastal freshwater inflow. This is reported to delay the formation of super polynyas in the model, but not to prevent them completely (Held et al., 2019). One reason for this partial success only may be that a realistic representation of the Antarctic Slope Current and related Antarctic Slope Front restricts the lateral spread of freshwater away from the shelf, as Lockwood et al. (2021) found for the GFDL-CM2.6 climate model with a comparatively high horizontal resolution."*

- p.20, l.459-460: I feel that Figure 11 could be moved to the Appendix as it is not a major figure of the paper.
changed

- p.21, l.470-472: You might find Adcroft et al. (2019) useful for this part of the discussion (e.g. see their Section 3.8.3). The introduction of the new hybrid coordinate system in MOM6 did not solve as much as hoped.
*"The Modular Ocean Model (MOM) introduced isopycnal coordinates in its latest version MOM6. Since several models are based on MOM, e.g. the ACCESS and the BCC models, we expect an improvement in deep water formation, Southern Ocean deep water properties and OWPs when these models successively upgrade to MOM6. Even though isopycnal coordinates were demonstrated to significantly reduce model drift and ocean heat uptake (Adcroft et al. 2019), they also have poor vertical resolution in weakly stratified regions such as the Weddell Sea and can not avoid spurious deep convection in all cases. Only in combination with adequate implementation of shelf processes and dense water overflows, optimal results could be obtained (Adcroft et al. 2019)."*

**Conclusion**
- p.21, l.477: "best agreement" with observations?
Yes, with observations. We added this clarification.
- p.21, l.485: Please remove the reference to the section.
→ removed as suggested

- p.21, l.485: Please specify what you refer to by "upwelling" (upward mixing of water resulting from gravitational instability or something else)? I do not recall this was specifically mentioned/discussed in the results section (the word "upwelling" does not appear there).
We did not discuss upwelling much in the results section, so we agree it should not appear out of the blue in the conclusions. We removed it from that section.
- p.21, l.487-488: There is only one paragraph in Section 5.3 on the comparison between the two ACCESS models, so I do not think that sentence summarizes the main point of that section.
We changed this:
*"In Section 5.3 we discussed how the problem of spurious open ocean deep convection in many CMIP6 models may be related to the OWPs biases. Half of the models show an over representation of OWPs (Table 2), which is a known problem in the field (e.g. Held et al., 2019; Gutjahr et al., 2019; Sellar et al., 2019), likely caused by unrealistic deep convection events in the Southern Ocean (see Table A1 for references). We have described some of the models' strategies to prevent open ocean polynyas and their caveats."*

**Data availability**
- Please provide the link for SOCCOM.
added

**Figures**

Figure 1
- Please specify in the caption what grey color corresponds to.
Added

Figure 2
- I think you should add a dark blue color in your colorbar corresponding to open water.
Added
- The colorbar is discrete but we see shades of blue around the sea ice edge or polynya edges. I assume this is an effect of the filter. Please add a note in the caption.
The shades of blue that were present earlier was an underlying sea ice concentration plot that we used to sanity check the classification. It was unnecessary for the plot and not explained in the colorbar, so we just removed it.

Figure 4
- Please explain what the two colorbars each correspond to (The caption only says "polynya areas").
We added an explanation:
*"The red-to-yellow colours indicate the number of years where polynyas were detected by our algorithm within the 165 years of the historical model run for each grid cell. The blue colors show the average sea ice concentration (in %)."*
- Please explain what the grey color corresponds to. Open ocean north of the sea ice edge and ice shelves both appear in grey.
Added
- Perhaps adding a contour of the sea ice edge on panel j) would help visualize the issue described by the authors.
Added contour of 1% sea ice concentration to both panels

Figure 6
- Please add a reference to the appropriate section and equations in the third sentence of the caption.
We added the reference
- The fact that some models do not have daily output or equivalent thickness leads to unequal length of the x-axis across panels a) to c) which makes comparing models across the different panels difficult. To remedy that, you could keep the full list of models on the x-axis for all panels with no data for models which miss output. I believe that would make things clearer.
As suggested, we added the missing models on the x-label in grey color

Figure 8
- "open water polynya" → "OWP".
→ changed as suggested

Figure 9
- Which domain did you take for the Weddell Sea? Please provide the range of longitudes and latitudes.
We limited the Weddell Sea sector to the longitudes (65°W-30°E) and added this information to the caption. We limited the latitudes to the wide range of (55°S-80°S). This

latitude limitation's purpose is just to exclude the Northern Hemisphere and some sea ice covered bays in South America from interfering with our algorithm.

- The labels of months on the x-axis are really packed. Maybe you could display one label every other label or rotate the labels?
We rotated the labels by 90° and that looks much better!
- I am not convinced that the use of symbols for displaying models is better than the use of plain and dashed lines with different colors. Symbols make the shape of the plot hard to grasp.
Actually, we used solid and dashed lines first. However, as one of us is colorblind, we found it hard to differentiate the resulting 14 colors, and confusing if dashed lines are overlapping solid lines in the same color.

Figure 10
- Please indicate in the caption the change of vertical scale between the first two and last two columns.
Added
- There are some ".8" and "-0.4" here and there on the figure that should be removed I believe.
Agree, removed.
- Does the T-S diagram use in situ or potential temperature? Which flavour of the density is used? Please specify.
Potential temperature and potential density are used. We added this in the description.
- Please specify the number of profiles for the float inside and outside the polynya (in the caption or in the text). It does not seem that there are many profiles inside the polynya.
This is true, only two profiles from within the polynya and 19 from under the sea ice. We added this to the caption.
- I suggest cutting the profiles at 2 km for the models to ease the comparison with the observations. The details of the Argo float profiles are hard to see and the most interesting features appear over the top 1 km for both models and observations anyway.
We discuss the depth of convection in the models, which reaches depths of up to 4000 m. Moreover,  inspired by your comment in the results section we added EN4 climatology data to the observational profiles, so that a meaningful comparison is possible even below 2000 m depth.

Table 2
- I think keeping three significant digits for all variables (especially for sa tot that could be written in 10 6 km 2 ) is enough. Also, that will make the numbers easier to read and compare.
We agree that the table looked cluttered and that sa_tot had too many digits. We rounded it to km² which improves the situation. However, we prefer to keep the area estimations with the same exponent for all columns, which we consider easier to compare.

Figure A1
- The figure is of poor resolution. Could you please try to improve that?
- I suggest you color-code the numbers in the legend (e.g. red for the coastal polynyas). That will be easier to read.
Changed to higher resolution, color coded legend

Figure A2
- Same comments as in Fig. A1.
Changed to higher resolution, color coded legend

**2. Additional changes**
Changes that are not listed in this document can be found in our response to the anonymous reviewer #1. Moreover, we received a helpful suggestion aside from the public review process, which we also want to address here. A reader of our preprint pointed out that we counted the total number of CMIP6 models in Table 2 of the cited Beadling et al. 2020 paper incorrectly, and that it would be preferable to stress which results were significant. Thank you for the corrections, we improved this part:

L436-438
Before: "However, Beadling et al. (2020) found that 30 of 35 CMIP6 models underestimated the ACC, 34 of 38 showed their wind stress curl minimum not sufficiently south and 33 of 38 underestimated the wind stress curl maximum. All these parameters are positively correlated with OWP activity (Campbell et al., 2019)"

Improved: "However, Beadling et al. (2020) found that *out of the 34 CMIP6 models they analysed, 29 underestimated the ACC (of which only 12 significantly), 30 show their wind stress curl not sufficiently south (5 significantly) and 30* underestimated the WSC minimum (9 significantly)."

**References**
Adcroft, A., Anderson, W., Balaji, V., Blanton, C., Bushuk, M., Dufour, C. O., et al. (2019). The GFDL global ocean and sea ice model OM4.0: Model description and simulation features. Journal of Advances in Modeling Earth Systems, 11, 3167-3211. https://doi.org/10.1029/2019MS001726´.
Heuzé, C., Ridley, J. K., Calvert, D., Stevens, D. P., and Heywood, K. J.: Increasing vertical mixing to reduce Southern Ocean deep convection in NEMO3.4, Geosci. Model Dev., 8, 3119-3130, https://doi.org/10.5194/gmd-8-3119-2015, 2015.
Kjellsson, J., and Coauthors, 2015: Model sensitivity of the Weddell and Ross Seas, Antarctica, to vertical mixing and freshwater forcing. Ocean Modell., 94, 141-â152, doi:10.1016/j.ocemod.2015.08.003.
Kurtakoti, P., Veneziani, M., Stössel, A., & Weijer, W. (2018). Preconditioning and Formation of Maud Rise Polynyas in a High-Resolution Earth System Model, Journal of Climate, 31(23), 9659-9678. https://journals.ametsoc.org/view/journals/clim/31/23/jcli-d-18-0392.1.xml.
Lockwood, J. W., Dufour, C. O., Griffies, S. M., & Winton, M. (2021). On the Role of the Antarctic Slope Front on the Occurrence of the Weddell Sea Polynya under Climate Change, Journal of Climate, 34(7), 2529-2548. https://journals.ametsoc.org/view/journals/clim/34/7/JCLI-D-20-0069.1.xml.
Stössel, A., D. Notz, F. A. Haumann, H. Haak, J. Jungclaus, and U. Mikolajewicz, 2015: Controlling high-latitude Southern Ocean convection in climate models. Ocean Modell., 86, 58-75, doi:10.1016/j.ocemod.2014.11.008. https://doi.org/10.1016/j.ocemod.2014.11.008.

**Referenced in our response (for further references see manuscript):**

- Beadling, R., Russell, J., Stouffer, R., Mazloff, M., Talley, L., Goodman, P., Sallée, J., Hewitt, H., Hyder, P., and Pandde, A.: Representation of Southern Ocean Properties across Coupled Model Intercomparison Project Generations: CMIP3 to CMIP6, Journal of Climate, 33, 6555–6581, 2020.
- Griffies, S. (2015). A handbook for the GFDL CM2-0 model suite.
- Hasumi, Hiroyasu. "CCSR ocean component model (COCO)." (Version 4.0) (2015).
- Heuzé, C.: Antarctic Bottom Water and North Atlantic Deep Water in CMIP6 models, Ocean Science, 17, 59–90, 2021.
- Keen, Ann, et al. "An inter-comparison of the mass budget of the Arctic sea ice in CMIP6 models." *The Cryosphere* 15.2 (2021): 951-982.
- Madec, G., Bourdallé-Badie, R., Bouttier, P. A., Bricaud, C., Bruciaferri, D., Calvert, D., ... & Vancoppenolle, M. (2017). NEMO ocean engine.
- Jungclaus, J. H., et al. "Characteristics of the ocean simulations in the Max Planck Institute Ocean Model (MPIOM) the ocean component of the MPI-Earth system model." *Journal of Advances in Modeling Earth Systems* 5.2 (2013): 422-446.

---

## Referee Report (RR1)

**Southern Ocean polynyas in CMIP6 models**

submitted to the Cryosphere by Martin Mohrmann et al.

I thank the authors for providing thorough and detailed responses to all my questions and for addressing all my major and minor comments. I find the revised manuscript ready for publication. Upon reading through the document with tracked changes, I just ran into a few typos and minor issues (listed below) that authors might want to consider.

**Minor or technical comments**

- Table 1: I believe that the GFDL CM4 uses the AM4.0 atmospheric model. The manuscript has been corrected to show "GFDL-OM4.0". OM4.0 refers to the ocean model. Please verify the information in Held et al. (2019). Same comment for the GFDL ESM4.

- l. 209: km should be $km^2$.

- l. 366: km should be $km^2$.

- Figure 5: The resolution of the figure is poor.

- l. 593: "polyny" → "polynya".

- l. 587: It is unclear to me what this factor is about. Also, there needs to be a "a" before "factor".

- Figure 10: In the caption, "ARGO" should be "Argo". Panel m, legend: There should also be ENA4.